# Dynamic control of decision and movement speed in the human basal ganglia

**Damian M. Herz** [1,2] ✉**, Manuel Bange**[2]**, Gabriel Gonzalez-Escamilla** [2]**, Miriam Auer**[2]**, Keyoumars Ashkan**[3]**, Petra Fischer** [4]**, Huiling Tan**[1]**, Rafal Bogacz**[1]**, Muthuraman Muthuraman** [2]**, Sergiu Groppa** [2] **& Peter Brown** [1]

To optimally adjust our behavior to changing environments we need to both adjust the speed of our decisions and movements. Yet little is known about the extent to which these processes are controlled by common or separate mechanisms. Furthermore, while previous evidence from computational models and empirical studies suggests that the basal ganglia play an important role during adjustments of decision-making, it remains unclear how this is implemented. Leveraging the opportunity to directly access the subthalamic nucleus of the basal ganglia in humans undergoing deep brain stimulation surgery, we here combine invasive electrophysiological recordings, electrical stimulation and computational modelling of perceptual decision-making. We demonstrate that, while similarities between subthalamic control of decision- and movement speed exist, the causal contribution of the subthalamic nucleus to these processes can be disentangled. Our results show that the basal ganglia independently control the speed of decisions and movement for each hemisphere during adaptive behavior.

Decision-making and motor control are often viewed as distinct processes, but in everyday choices the two are inextricably intertwined. For example, we devalue options that are deemed too effortful to obtain and adapt both the speed of deliberation and of our movements so as to achieve goals as soon as possible[1,2]. Even though relatively little is known about common neural mechanisms underlying decision-making and movement control, the basal ganglia are thought to represent a crucial interface between the two[1,3,4]. In particular, the subthalamic nucleus (STN) is thought to be a central hub in determining when deliberation should be terminated and how vigorously a movement should be performed[5,6], as it receives afferents from a broad spectrum of cortical areas involved in decision-making and movement control[3,7]. A prevalent hypothesis is that the STN can hold a motor response until sufficient evidence is collected or any response conflict has been resolved thus contributing to response selection and decision speed[8–10]. A separate line of research has indicated a dominant role of the basal ganglia in modulating movement vigor, in particular movement speed, determining how quickly a movement should be performed to indicate the choice[1,11–13]. These hypothesized roles of the basal ganglia and STN share many commonalities in that they together determine the time it takes to reach a goal given its expected value[4] or time pressure[14], but are typically studied in separate fields (decision-making and motor control).

It remains unclear to what extent these putative functions of the STN are controlled separately or by a common signal[15]. To address this question, we here recorded STN local field potentials (LFP) and applied bursts of electrical STN stimulation in patients with Parkinson's disease (PD) whilst assessing their ability to make decisions and perform movements. Based on previous correlative evidence[16–18] we hypothesized that the correlates and causal contributions of STN to decision speed precede its effect on movement speed.

[1]MRC Brain Network Dynamics Unit at the University of Oxford, Nuffield Department of Clinical Neurosciences, University of Oxford, Oxford, United Kingdom. [2]Section of Movement Disorders and Neurostimulation, Department of Neurology, Focus Program Translational Neuroscience (FTN), University Medical Center of the Johannes Gutenberg-University Mainz, Mainz, Germany. [3]Department of Neurosurgery, King's College Hospital, London, United Kingdom. [4]School of Physiology, Pharmacology and Neuroscience, University of Bristol, Bristol, United Kingdom. ✉e-mail: damian.m.herz@gmail.com

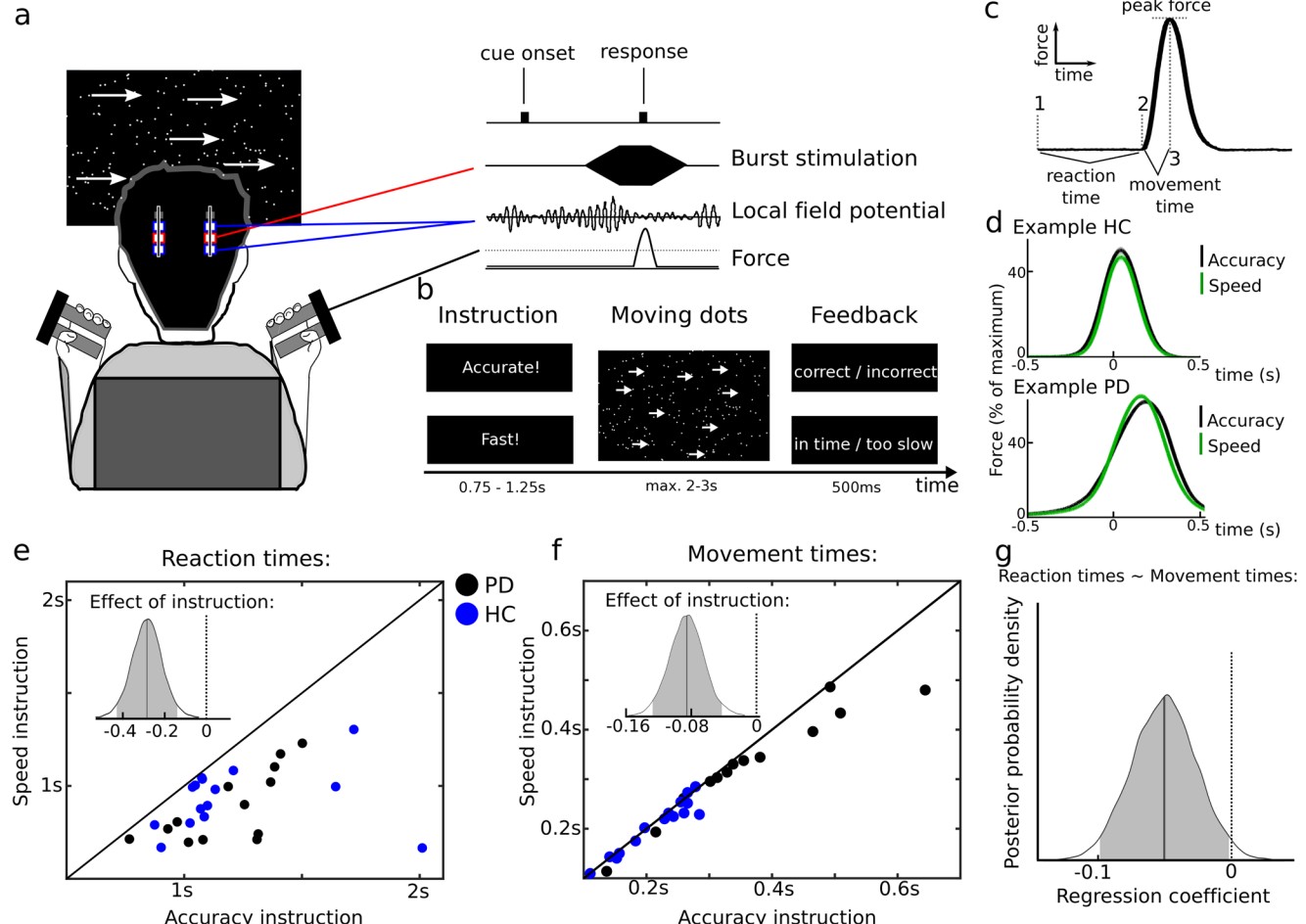

**Fig. 1 | Experimental setup and behavioral results. a** During the task people indicated whether a cloud of moving dots (8% coherence) moved to the left or right by pressing a dynamometer with their left vs. right hand. In patients with Parkinson's disease (PD) electrode extension cables from the subthalamic nucleus enabled both recordings of local field potentials and stimulation. **B** During the task patients were instructed to respond as fast (deadline: 2s) or as accurately (deadline: 3s) as possible at each trial in randomized order. **C** Force recordings allowed the subdivision of response times into reaction times (from onset of moving dots cue (1) to onset of movement (2)) and movement time (from 2 to peak force (3)). **D** Examples of trial-averaged force recordings are shown separately for speed (green) and accuracy (black) trials for one healthy control (HC) participant and one PD patient. **E** Single participant mean reaction times for speed (*y* axis) vs. accuracy trials (*x* axis). The posterior density of the effect of instruction is shown in the inset. PD patients (*n* = 13) are shown as black circles, HC participants (*n* = 15) as blue circles. **F** Same as E but for movement times. **G** Posterior density of the regression coefficient of movement times on reaction times. Source data are provided as a Source Data file.

Moreover, we aimed to elucidate the conceptual role of STN during decision-making. One possibility is that the STN contributes to the computation of a global decision-threshold construct (i.e., affecting all movement) determining the agent's general level of cautiousness putatively through its connections with the prefrontal cortex[5,19,20]. Another possibility is that the STN is involved in setting decision thresholds at a lower hierarchical level, controlling decision speed for each hemisphere[21,22]. To test these possibilities, we applied bursts of STN stimulation to, respectively, both hemispheres and only one hemisphere at a time in separate sessions allowing us to assess whether the functional lateralization of STN does not only apply to control of movement[17,23–25], but also to decision-making.

Here, we show that the STN can independently control movement and decision speed in distinct processing windows and that it contributes to setting decision thresholds for each hemisphere during adaptive behavior.

## Results

### People speed up decisions and movement during time pressure

Thirteen PD patients implanted with deep brain stimulation (DBS) electrodes (clinical details are listed in supplementary table 1) and 15 healthy, age-matched control participants (HC) performed a perceptual decision-making task. Participants had to indicate whether they perceived that a cloud of moving dots was moving to the left or right by pressing a dynamometer with their left or right hand after being instructed to respond as fast (speed instruction) or as accurately (accuracy instruction) as possible at each trial, see Fig. 1A, B. By recording force from the dynamometer, response times could be separated into reaction times (cue to movement onset) and movement times (movement onset to peak force), see Fig. 1C, and these were compared between speed and accuracy instructions (Fig. 1D). First, we compared the behavior of the PD group, studied on dopaminergic medication, and the HC group in detail to address the generalizability of the results before relating the behavioral adjustments to STN activity.

We found that speed instructions significantly decreased reaction times compared to accuracy instructions (95% Bayesian Credible Interval (CrI) [−0.429:−0.139]) irrespective of Group (main effect of Group, CrI [−0.101:+0.249]; Instruction*Group interaction (IA), CrI [−0.152:+0.235]), see Fig. 1E. Similarly, movements were significantly faster after speed compared to accuracy instructions (CrI [−0.127:−0.043]), see Fig. 1F. As expected movements were also

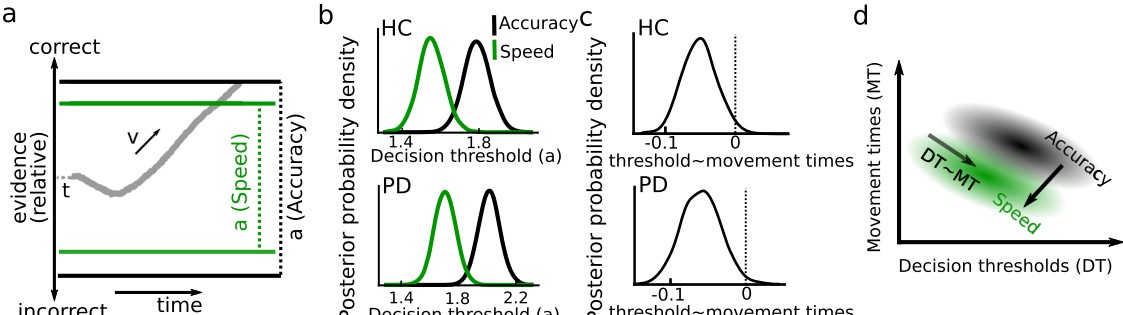

**Fig. 2 | Hierarchical drift-diffusion modeling (HDDM). a** The main parameters of the model are the non-decision time $t$, the drift rate $v$, and the decision threshold $a$ determining when sampled sensory evidence (gray trace) results in a choice that is correct (upper boundary) or incorrect (lower boundary). Faster decision during speed pressure can be implemented through a reduction of $a$ (shown in green). **B** Both Parkinson's disease (PD) patients and healthy controls (HC) had lower decision thresholds after speed compared to accuracy trials. **C** Posterior density for regression of movement times on decision thresholds for HC and PD. **D** Schematic representation of the main findings from the behavioral and computational analyses showing the intrinsic relationship between movement times and decision thresholds (arrow marked DT-MT) as well as their shift to lower values after speed instructions (arrow from accuracy to speed). The black and green clouds represent distributions of movement times and decision thresholds during, respectively, accuracy and speed instructions.

significantly faster in healthy people compared to PD patients (CrI [−0.717:−0.189]), but there was no Instruction*Group IA (CrI [−0.011:+0.100]) indicating that both groups adjusted their movements to a similar extent. Similar to our previous observations[16] PD patients had lower accuracy rates compared to healthy people (~65% vs. ~75%, change in log-odds CrI [+0.068:+1.051]). Importantly there was no Instruction*Group IA (CrI [−0.411:+0.299]), i.e., accuracy rates were not differently affected by instructions in PD patients and healthy controls, and all participants performed above chance level. Accuracy rates were not significantly different between speed and accuracy instructions (CrI [−0.349:+0.176]), but participants committed errors disproportionately faster after speed vs. accuracy instruction as indicated by a significant Instruction*response accuracy IA on reaction times (CrI [+0.042:+0.223]), which did not differ between groups. Notably, relatively faster errors after speed instructions indicate that these were due to lower levels of evidence at the time of the choice rather than higher levels of sensory noise[26]. There were no effects of Instruction or Group on peak force. Furthermore, we did not find differences in movement times depending on the response side (left vs. right) or response accuracy, nor did movement times increase with the duration of the experiment indicating that the participants were not affected by fatigue during the short task (see supplementary table 2 for the 95% Bayesian CrI and non-Bayesian confidence intervals (CI), t- and p-values of all statistical tests, supplementary table 3 for results when only including correct trials and supplementary table 4 when using non-log-transformed single-trial data).

We then asked whether the observed changes in reaction and movement times were related to each other using movement times at each trial as the independent variable and reaction times as the dependent variable in a regression analysis. This analysis showed a significantly negative slope, i.e., faster movements were related to longer reaction times (CrI [−0.099:−0.002]), see Fig. 1G. This relationship was not affected by Group or Instruction (Movement time*Group IA CrI [−0.050:+0.080], Movement time*Instruction IA CrI [−0.052:+0.034], Movement time*Group*Instruction IA CrI [−0.019:+0.087]).

Together, these findings revealed that participants both decreased their reaction and movement times during time pressure and that these changes were similar in PD patients and healthy people. Furthermore, at the single-trial level longer reaction times were related to faster movements and this relationship did not differ depending on the group or instructions. Next, we attempted to gain a better understanding of these findings by modeling the latent changes in decision-making parameters underlying the observed behavior.

## People speed up decisions by lowering the evidence required for a choice

We used a well-known, parsimonious model termed the drift-diffusion model (DDM) that has been successfully applied to a range of decision-making tasks[26] and contains three main parameters: the decision threshold, which determines how much evidence people require before committing to a choice, the drift rate reflecting the rate of evidence accumulation and the non-decision time incorporating processes not directly related to decision-making (such as afferent delay or motor preparation, see Fig. 2A and methods for more details). Including an additional parameter reflecting the starting point of evidence accumulation (bias) did not alter any of the reported results. Using a Bayesian hierarchical DDM (HDDM)[27] we first fitted a model allowing modulations of all parameters by Instruction and Group. We found that speed vs. accuracy instructions significantly reduced decision thresholds (CrI [−0.398:−0.119]), see Fig. 2B, but did not affect drift rates (CrI [−0.251:+0.238]) or non-decision times (CrI [−0.128:+0.053]). There were main effects of Group on decision thresholds (PD > HC, CrI [+0.054:+0.324]), drift rates (PD < HC, CrI [−0.622:−0.131]), and non-decision times (PD < HC, CrI [−0.415:−0.238]). Importantly, however, there were no IA of Group*-Instruction on any of the parameters (decision thresholds CrI [−0.346:+0.199], drift rates CrI [−0.479:+0.496], non-decision times CrI [−0.104:+0.259]) showing that PD patients adjusted their decision-making parameters based on instructions similarly to healthy people, as already suggested by the behavioral analysis.

In a next step, we aimed to assess whether modulations of distinct decision-making parameters could explain the observed trial-by-trial relationship between movement and reaction times. To this end we first constructed a model that only contained the significant Instruction effects from the previous model (i.e., on decision thresholds) for each group. This simple model predicted the observed data of both PD patients and healthy people well (supplementary Fig. 1). Then we included single-trial estimates of movement times as independent variable in the HDDM regression model. In both groups, we found a significant negative relationship between movement times and decision thresholds (HC CrI [−0.094:−0.004], PD CrI [−0.110:−0.011]), see Fig. 2C, but no IA with Instruction (HC CrI [−0.010:+0.100], PD CrI: [−0.035:+0.115]), nor any relationship with drift rates (HC CrI [−0.022:+0.079], PD CrI [−0.089:+0.022]) or non-decision times (HC CrI [−0.005:+0.010], PD CrI [−0.002:+0.003]).

In summary, we found that the observed reduction in reaction time during speed vs. accuracy instructions could be parsimoniously explained by a decrease in the decision threshold allowing responses at lower levels of sensory evidence. Trial-by-trial variations in

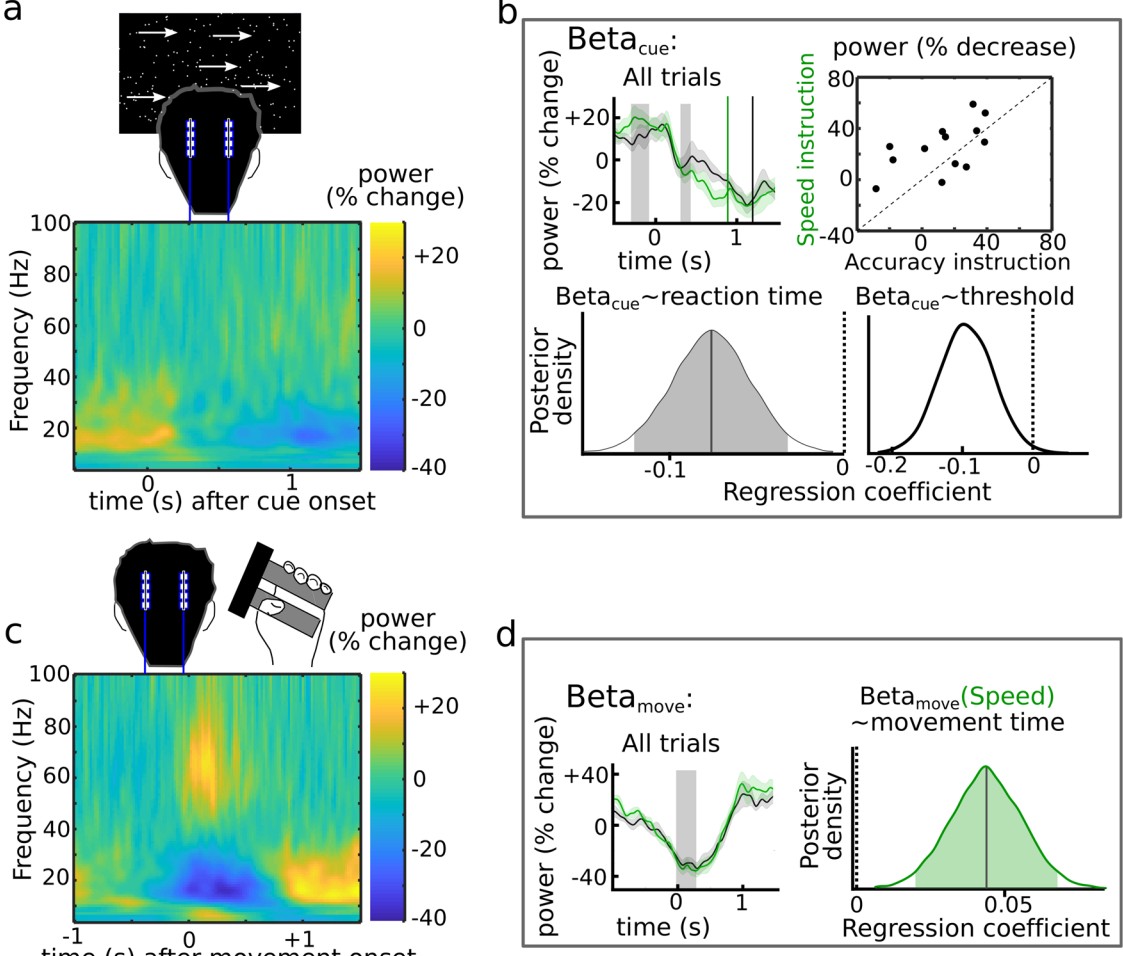

**Fig. 3 | Local field potential recordings from subthalamic nucleus. a** Grand average for spectra aligned to the moving dots cue. **B** Results for cue-aligned changes in beta (–13–30 Hz) power (Beta$_{cue}$) measured as change in beta power from precue (300–100 ms precue) to post cue (320–400 ms post cue) across participants (n = 13). **C** Grand average for spectra aligned to movement onset. **D** Results for movement-related (0–300 ms post-movement) changes in beta power (Beta$_{move}$) across participants (n = 13). In **B** and **D** shaded areas around % power change (green: speed; black: accuracy) represent SEM and gray vertical boxed indicate time windows from which power was extracted. Green and black vertical lines in **B** indicate mean reaction time for speed and accuracy trials. Source data are provided as a Source Data file.

movement times were specifically related to changes in the decision threshold, while there was no relationship with changes in drift rates or non-decision times. This suggests that when people required more evidence for their choice, they then indicated this choice by a faster movement. Speed vs. accuracy instruction did not alter this relationship (i.e., the slope), but shifted it toward overall lower decision thresholds and shorter movement times, as schematically illustrated in Fig. 2D.

## STN reflects adjustments of decision- and movement speed in distinct windows

During the task we recorded LFP directly from the STN through temporarily externalized DBS electrodes in PD patients. This allowed us to analyze whether STN activity changes were related to the observed adjustments in decision time, movement time, or both.

After onset of the moving dots cue the strongest changes in STN activity were observed in the beta band (–13–30 Hz) which showed a sharp reduction in power immediately after the cue (Fig. 3A) and was lower in the contralateral compared to the ipsilateral hemisphere (supplementary Fig. 2A). Across trials beta power after the onset of the moving dots cue (Beta$_{cue}$) decreased more strongly after speed compared to accuracy instructions (CI [+0.006:+0.246], P = 0.041). This remained significant when excluding any motor responses that

fell into this time window (CI [+0.015:+0.266], P = 0.031) and when comparing beta at the trough after the cue rather than computing the pre- vs. post cue difference (CI [–0.002:–0.212], P = 0.046), demonstrating that the stronger beta power drop after speed instructions was neither driven by early movement responses nor baseline differences in the pre-cue period. At the single-trial level a stronger decrease in beta power was predictive of shorter reaction times (CrI [–0.120:–0.033]), but not of changes in movement times (CrI [–0.014:+0.027]). These results were robust to adjusting the exact baseline period used for extracting beta power (supplementary table 5). Entering single-trial values of Beta$_{cue}$ into HDDM revealed a significant negative relationship with decision thresholds (CrI [–0.162:–0.020]); the stronger Beta$_{cue}$ decreased the lower were decision thresholds, irrespective of Instruction (Beta$_{cue}$*Instruction IA CrI [–0.047:+0.142]). It was not related to changes in drift rate (CrI [–0.087:+0.053]) or non-decision times (CrI [–0.005:+0.002]). These results are illustrated in Fig. 3B.

Next, we analyzed changes in STN activity in relation to movement onset where there was a strong decrease in STN beta activity (Fig. 3C), which was lower in the contralateral compared to the ipsilateral hemisphere (supplementary Fig. 2B). Beta power during the movement (Beta$_{move}$) was slightly lower during speed compared to accuracy instructions, but this did not reach significance (CI [–0.073:+0.010],

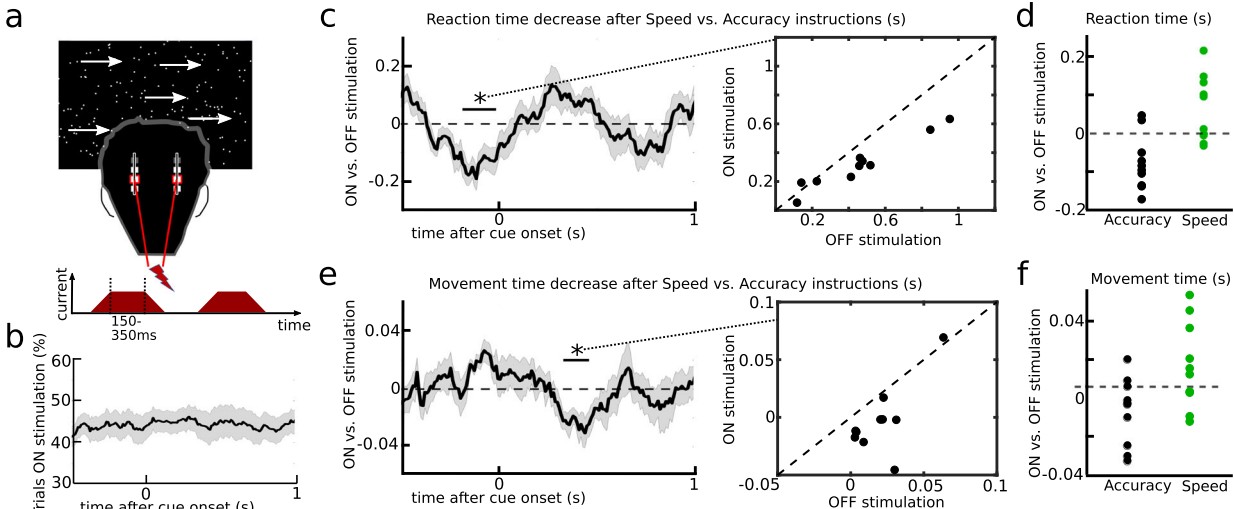

**Fig. 4 | Behavioral effects of bilateral burst stimulation. a** Bursts were given at random time points throughout the experiment. **B** This resulted in stimulation occurring on ~50% of trials for any 100 ms moving time window. **C** Stimulation impaired patients' ability to adjust reaction times in a time window confined to 180-10 ms precue (DBS$_{RT}$), which is marked by an '*' (corrected for multiple comparisons using cluster-based permutation tests and two-sided alpha-level of 0.05). Single-subject responses ($n = 10$) to DBS$_{RT}$ are shown in the panel to the right. **D** DBS$_{RT}$ effects shown for each participant ($n = 10$) separately for accuracy (black)

and speed trials (green). **E** Stimulation impaired patients' ability to adjust movement times in a separate time window (330–460 ms post cue, DBS$_{MT}$) marked by an '*' (corrected for multiple comparisons using cluster-based permutation tests and two-sided alpha-level of 0.05). Single-subject responses ($n = 10$) to DBS$_{MT}$ are shown in the panel to the right. **F** DBS$_{MT}$ effects shown for each participant ($n = 10$) separately for accuracy (black) and speed trials (green). Shaded areas in **B**, **C** and **E** represent SEM. Source data are provided as a Source Data file.

$P = 0.127$). However, its by-trial effect on movement times differed after speed vs. accuracy instructions (CrI IA [+0.010:+0.069], CrI of main effect [−0.015:+0.026]). Post hoc tests revealed that lower single-trial values of Beta$_{move}$ were predictive of shorter movement times after speed instructions (CrI [+0.020:+0.068]), see Fig. 3D, but not after accuracy instructions (CrI [−0.018:+0.025]). Changes in Beta$_{move}$ were not related to changes in reaction times (CrI of main effect [−0.033:+0.051], CrI Beta$_{move}$*Instruction IA [−0.089:+0.033]).

Finally, based on previous studies[16–18,20,28], we analyzed cue-aligned theta (4–8 Hz) power and movement-aligned gamma (55–80 Hz) power. Replicating these previous studies we found that theta power correlated with reaction times and decision thresholds depending on instructions, but not with movement times, and that movement-related gamma power correlated with movement speed, but not reaction times (supplementary Fig. 3).

In summary, we found that STN beta activity was related to both adjustments of reaction times and movement during speed vs. accuracy instructions. Beta power after the cue decreased more strongly after speed instructions and trials with a stronger decrease were related to shorter reaction times and lower decision thresholds, but not movement speed. In contrast, beta power during the movement predicted faster movements when speed was emphasized, but had no relationship with reaction times. Thus, modulation of beta power reflected the shift to lower decision thresholds and faster movements when speed was emphasized, but over different time windows. The temporal independence of this modulation of reaction times and movement by STN beta power was further corroborated by the lack of correlation between Beta$_{cue}$ and Beta$_{move}$ (CrI [−0.099:+0.032]).

Next, to assess whether the STN is causally involved in adjustments of decision- and movement speed we applied bursts of stimulation to the STN during the same experimental task in a second session.

**STN causally controls decision- and movement speed in separate windows**
To disentangle timing-specific effects of STN on the observed behavioral adjustments we applied short bursts of clinically effective

DBS (mean duration: 250 ms, see Fig. 4A) to STN in both hemispheres, which were ramped up and down (ramping was defined as no stimulation, see methods). Burst duration and interval were controlled so that for any given 100 ms time window during the task stimulation was applied in ~50% of trials (Fig. 4B). We then compared trials with and without stimulation regarding their effects on adjustments of reaction times and movement using a sliding window approach and performed cluster-based permutation tests to correct for multiple comparisons across all time windows (see methods for more details).

We found that stimulation impaired reaction time adjustments between speed vs. accuracy trials in a temporally confined window from 180-10 ms before cue onset ($P_{cluster} < 0.05$), which we hereafter refer to as DBS$_{RT}$ (see Fig. 4C for group and single subject effects). DBS in this time window did not significantly alter accuracy rates (CI [−0.093:+0.111], $P = 0.848$). Post hoc tests showed that stimulation reduced reaction times during accuracy trials (CI [−0.119:−0.025], $P_{corrected} = 0.014$), while the increase in reaction times during speed trials did not reach significance (CI [+0.003:+0.126], $P_{corrected} = 0.084$), see Fig. 4D. This was confirmed by HDDM showing a significant reduction of decision thresholds in accuracy trials (CrI$_{90}$ [−0.243:−0.003]), while the change was not significant in speed trials (CrI$_{90}$ [−0.110:+0.088]). In a control analysis, we defined incremental periods of the ramping period as stimulation, since it is conceivable that DBS might have effects on reaction time adjustments at intensities that are lower than clinically effective stimulation, i.e., during ramping. This analysis demonstrated that effects of DBS$_{RT}$ were robust to changes in the exact definition of effective stimulation (supplementary table 6). DBS$_{RT}$ also remained significant when only including correct trials (CI [−0.260:−0.035], $P = 0.016$).

Movement speed adjustments were affected by stimulation in a different time window occurring later, from 330–460 ms after cue onset ($P_{cluster} < 0.05$, Fig. 4E), which we hereafter refer to as DBS$_{MT}$. Post hoc tests showed that this was mainly driven by an effect of stimulation on movement speed after accuracy instructions reducing movement times (Fig. 4F), but the post hoc effects did not reach significance after correction for multiple comparisons in either

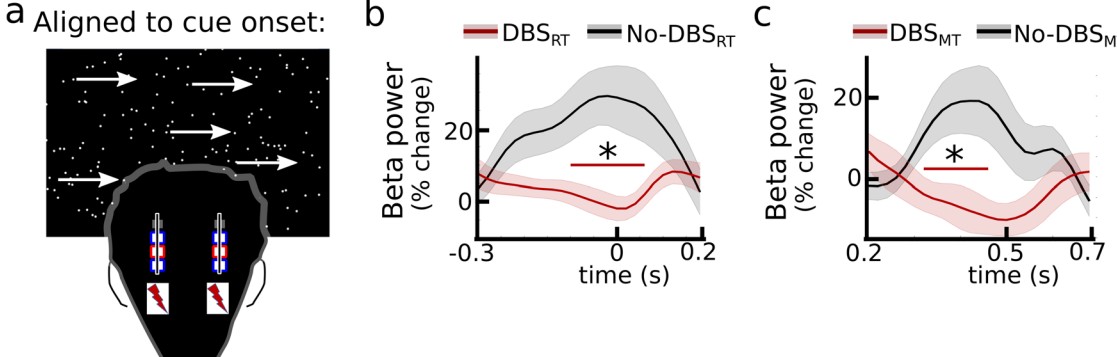

**Fig. 5 | Local field potential recordings during stimulation. a** Beta power is aligned to cue onset. **B** Cue-aligned beta power for trials where stimulation was applied during DBS_{RT} (red) compared to trials where stimulation was not applied in DBS_{RT} (termed No-DBS_{RT}, black) averaged across participants (n = 10). Significant differences according to cluster-based permutation tests are marked by red bars

with a *. **C** Same as **B**, but for DBS_{MT} vs. No-DBS_{MT}. Note that the shown beta traces are not affected by movement-related beta power reductions and that the data were normalized to the mean across the whole recording (see methods). Shaded areas represent SEM.

condition (Accuracy: CI [−0.026:−0.001], $P_{corrected}$ = 0.083; speed: CI [−0.006:+0.027], $P_{corrected}$ = 0.338).

Stimulation effects on reaction and movement times, respectively, in the two different time windows were not correlated across subjects (rho = −0.042, P = 0.908, Pearson correlation). We did not find any effects of stimulation when aligning the data to movement onset (supplementary Fig. 4).

Together, these results demonstrate that the STN is causally involved in the observed behavioral adjustments and that changes in reaction and movement times are controlled independently in distinct temporal windows. These two-time windows are similar to those in which STN beta power changes were associated with adjustments in reaction and movement times. To assess more directly whether the applied bursts of STN stimulation affected the dynamical modulation of beta power[29–32], we compared task-related changes in STN beta power on and off stimulation.

## STN stimulation interferes with dynamic changes in beta activity

During stimulation, we simultaneously recorded STN activity from bipolar contacts surrounding the stimulation electrode. Using common-mode rejection, filtering and artifact removal (see methods for more details), we were able to recover the dynamic changes in beta power observed without stimulation across all trials (i.e., reduction in beta power after the cue and during the movement), see Supplementary Figs. 5, 6. Aligning beta power to stimulation onset showed a strong stimulation-related reduction in beta power (supplementary Fig. 7) as expected from the previous studies[29–32]. To analyze whether DBS_{RT} and DBS_{MT} also were accompanied by stimulation-induced beta power reductions, we compared trials with stimulation vs. no stimulation in the two-time windows aligned to cue onset (Fig. 5A, see methods for more details). DBS_{RT} reduced beta power from 100 ms pre cue to 60 ms post cue ($P_{cluster}$ < 0.05), see Fig. 5B. DBS_{MT} reduced beta power from 320–460 ms post cue ($P_{cluster}$ < 0.05), see Fig. 5C. Thus, the timing-specific behavioral effects of stimulation were accompanied by reductions in STN beta power.

While the neural control of movement in the basal ganglia is well-known to show strong lateralization[17,18,23–25], how the basal ganglia control decision speed is less clear. To assess whether STN control of decision thresholds is global (i.e., affects both body sides) or the STN controls decision speed mainly for the contralateral body side similar to its effect on movement, we applied bursts of STN stimulation to one hemisphere at a time during the final part of the study.

## STN controls decision speed for each hemisphere

In a final session, patients performed the experimental task whilst STN stimulation was applied unilaterally (left and right STN were stimulated in consecutive runs with counterbalanced order, see methods). This allowed us to distinguish trials where the STN contralateral to the moving hand was stimulated (Fig. 6A) from trials where the ipsilateral STN was stimulated (Fig. 6D).

When stimulating the STN contralateral to the moving hand we found a significant effect on reaction times in an almost identical time window to the significant behavioral effects of bilateral stimulation, i.e., 230–40 ms before cue onset ($P_{cluster}$ < 0.05, hereafter termed DBS_{contra}), see Fig. 6A, which significantly reduced decision thresholds (CrI [−0.302:−0.017], Fig. 6B). DBS_{contra} remained significant when only including correct trials (CI [−0.272:−0.035], P = 0.018). As for bilateral stimulation, we conducted a control analysis defining incremental periods of the ramping period as stimulation, which demonstrated that the effects of DBS_{contra} were robust to changes in the exact definition of effective stimulation (supplementary table 6). DBS_{contra} did not significantly affect accuracy rates (CI [−0.075:+0.102], P = 0.729) and, contrary to bilateral stimulation, the effect on reaction times did not differ between speed vs. accuracy trials (CI [−0.102:+0.270], P = 0.320).

There were no significant effects on reaction times when stimulating the STN ipsilateral to the effector ($P_{cluster}$ > 0.05, Fig. 6D) and ipsilateral stimulation did not affect decision thresholds (CrI [−0.199:+0.069], Fig. 6E). The effect of contralateral stimulation on reaction time was significantly stronger than ipsilateral stimulation in a direct comparison albeit only using a one-tailed test ($P_{one-tailed}$ = 0.042). Thus, unilateral stimulation increased decision speed by lowering the decision threshold only for the contralateral body side. This suggests that the behavioral effects during bilateral stimulation were mainly driven by the contralateral STN. If so, this would predict that across subjects, people who had stronger behavioral effects of bilateral stimulation also should show strong effects after contralateral stimulation and vice versa, while this should not hold true for ipsilateral stimulation. Of note, since the same contacts were stimulated during contra- and ipsilateral stimulation any differences could not be explained by differences in lead localization or stimulation efficacy. In line with our prediction, we found that the effect on reaction times during DBS_{contra} significantly predicted the behavioral effect we had observed during bilateral stimulation ($r^2$ = 0.63, $P_{corrected}$ = 0.037), see Fig. 6C, while this was not the case for ipsilateral stimulation ($r^2$ = 0.18, $P_{corrected}$ = 0.574), see Fig. 6F. The relationship between contra- and bilateral stimulation remained significant when using both unilateral

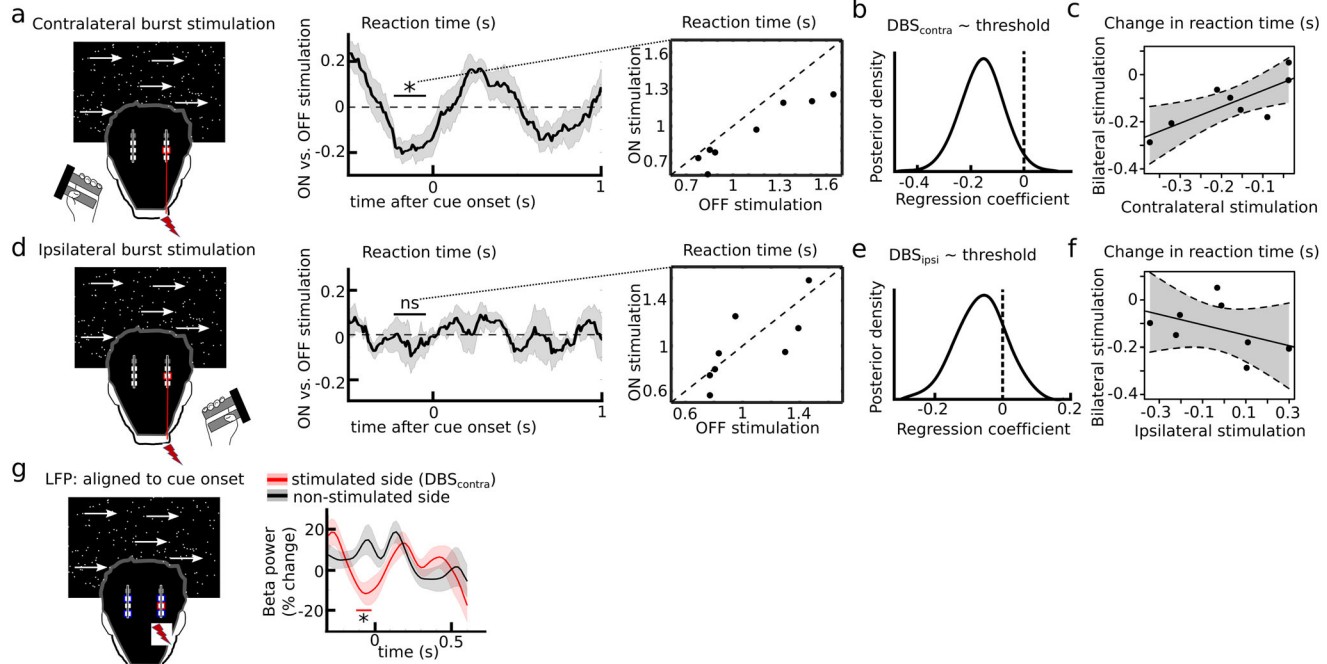

**Fig. 6 | Unilateral stimulation. a** Trials were divided into contralateral stimulation (e.g., left-handed response during stimulation of right STN) and ipsilateral stimulation (e.g., right-handed response during stimulation of right STN, shown in **D**). Contralateral stimulation reduced reaction times in a time window from 230–40 ms pre cue (compare to Fig. 4C). Single participant ($n = 8$) responses for this time window (DBS$_{contra}$) are shown in the right panel. **B** DBS$_{contra}$ significantly reduced decision thresholds. **C** Across subjects ($n = 8$), the behavioral effect of contralateral stimulation on reaction times predicted the effects of bilateral stimulation on reaction time changes (predicting ~67% variance, regression slope illustrated by black line, 95% CI shown by dotted lines). **D** During ipsilateral stimulation no significant effects on reaction times were observed across subjects ($n = 8$, corrected

for multiple comparisons using cluster-based permutation tests and two-sided alpha-level of 0.05). Single participant ($n = 8$) responses from the same time window as above, i.e., 230–40 ms precue (here termed DBS$_{ipsi}$) are shown in the right panel. **E** DBS$_{ipsi}$ did not affect decision thresholds. **F** Across subjects ($n = 8$), the behavioral effects of ipsilateral stimulation did not predict the effects of bilateral stimulation (regression slope illustrated by black line, 95% CI shown by dotted lines). **G** Cue-aligned changes in beta power for the stimulated STN (during DBS$_{contra}$, red) and the non-stimulated STN (black) averaged across participants ($n = 8$). The red bar with a * in the right panel indicates significant differences between stimulated and non-stimulated STN. Shaded areas in **A**, **D**, and **G** represent SEM. Source data are provided as a Source Data file.

stimulation conditions as predictors in a multiple regression model ($P = 0.043$).

In a final analysis, we assessed how STN beta power was altered by unilateral stimulation. We hypothesized that the stimulated STN, but not the non-stimulated STN, should show a similar reduction in beta power as observed during bilateral stimulation, since it was sufficient to yield behavioral changes. In line with this prediction, we found that timing-specific stimulation (DBS$_{contra}$) significantly reduced STN beta power compared to the non-stimulated hemisphere from 120 to 40 ms precue ($P_{cluster} < 0.05$), see Fig. 6G.

## Discussion

What determines the speed with which we think and act? We slow down movement and decision speed when we are uncertain about outcomes[33], move faster toward things that we value more[4], or when facing time pressure[14]. In this study, we reveal correlates of decision and movement speed in the STN, a central part of the basal ganglia that has long been thought to constitute a crucial interface between decision-making and movement control[1,3,4]. We demonstrate that STN activity, as reflected by oscillations in the beta band that mainly localize to the dorsolateral STN[24,34,35], can reflect both decision and movement time, but in temporally separate time windows and statistically independent from each other. We further demonstrate the mechanistic importance of STN activity changes by applying bursts of electrical stimulation to the STN. Burst stimulation affected adjustments of decision and movement speed but only when applied in critical, separate time windows. Thus, DBS$_{RT}$ reduced reaction time whilst simultaneously augmenting the initial task-dependent drop in

beta power. The results from trials without stimulation confirmed that stronger early drops in beta power were indeed associated with shorter reaction times and lower decision thresholds. In contrast, stimulation applied ~400 ms after the cue (DBS$_{MT}$) reduced movement time. This was accompanied by a correspondingly stronger suppression of beta power, over and above that seen in the unstimulated condition. The latter confirmed that lower beta power was associated with shorter movement times, albeit mostly in the speed condition.

Modulations of beta power during decision-making and motor control have also been reported at the cortical level in healthy participants[36–38]. Together with the observation that patients' behavioral adjustments during the task were similar to those of healthy people, this suggests that our results may generalize beyond the studied patient population even though we cannot conduct direct electrophysiological STN recordings in healthy people.

We also found a negative correlation between decision thresholds and movement time. The more evidence people accumulated the faster were their movements to indicate the choice irrespective of instructions. This is compatible with the observation that choices with higher levels of certainty are accompanied by faster movement[33]. However, this relationship could also be driven by better movement preparation during longer reaction times. The lack of a correlation between Beta$_{cue}$ and Beta$_{move}$ indicates that the context-independent relationship between decision thresholds and movement times might be related to other spectral components or neural circuits outside the STN.

In sum, our results confirm previous findings showing STN correlates of decision thresholds[16,20] and movement speed[17,18,23] and

significantly extend these by showing that they can be controlled independently allowing the STN, presumably together with its interconnected networks, to flexibly control behavior. While this enables the agent to adapt deliberation and movement together, e.g., for minimizing the time it takes to achieve a goal[1], it still retains the flexibility to reduce one but not the other, e.g., when deliberation time can be sacrificed during relatively easy decisions, but the goal can only be reached with intricate, slower movement[39]. More broadly our results are in line with the basal ganglia determining when to commit to a choice[8–10] (decision speed) and how vigorously to indicate the choice[1,11–13] (movement speed) in distinct processing windows.

Another key finding from this study is that, similar to the lateralized control of movement in the STN[17,23–25], its effect on decision speed is related to choices indicated with the contralateral, but not ipsilateral, hand. While our previous study demonstrated a causal contribution of STN to decision threshold adjustments according to by-trial changes in task difficulty[40], the current study confirmed that this also extends to decision threshold adjustments driven by trial-by-trial changes in time pressure and elucidated a strong lateralization of this effect. The latter is in line with recent evidence showing that motor cortex influences decision cautiousness of the contralateral body side in humans[22] and that optogenetic manipulation of the basal ganglia can bias perceptual decisions to the contralateral vs. ipsilateral side in rodents[21]. Interestingly, while bilateral stimulation affected reaction times depending on task context by mainly speeding up decisions during the accuracy condition, contralateral stimulation reduced reaction times irrespective of condition. This suggests that context-specific decision threshold adjustment might require the controlled activation of bilateral STN.

Together, the results suggest that the STN might primarily be involved in threshold adjustments at a lower hierarchical level. More precisely, while cortical computations might primarily determine a higher-order, "global" decision threshold determining how cautiously an agent generally (i.e., irrespective of the exact movement or body side) behaves depending on task context, risk and control demands[14,41], at the level of the basal ganglia this appears to be specific to each hemisphere (here controlling left vs. right hand movement).

There are some limitations to this study. First, due to the rapid ON-OFF cycling of burst stimulation subthalamic neurons might not have reached a stable baseline in the stimulation interval despite ramping down and the interval might thus constitute a post-stimulation rather than an off-stimulation condition[32,42]. Furthermore, we here titrated stimulation intensity to clinically effective stimulation, i.e., DBS intensities that resulted in motor improvement. However, it could be that DBS might already affect decision processes at lower intensities[43]. We addressed this to some extent by including incremental periods of the ramping period as stimulation showing that the results were robust to changing the exact definition of stimulation vs. no stimulation. Future studies could address these issues by focusing stimulation bursts on pre-defined windows of interest at each trial (leaving several seconds off stimulation) and assessing the effect of different stimulation intensities on cognitive and motor processes. Finally, we used a well-validated task for probing the speed-accuracy trade-off. However, it has been shown that this task evokes much stronger changes in reaction times than accuracy rates[16,26,40]. While analysis of error reaction times and computational modeling confirmed that the changes between conditions were best explained by a reduction of the decision threshold, future studies could use paradigms that are more sensitive to effects of decision speed on choice accuracy. It remains to be elucidated to what extent DBS-mediated interruptions of decision-making processes are related to impulsiveness, which is observed during conventional DBS in some PD patients[44,45] and might be avoidable using adaptive DBS approaches[46,47].

## Methods

### Sample size
Before conducting the study we recorded pilot data from 12 healthy people performing the same experimental paradigm (described below). Our main focus was the effect of speed vs. accuracy (SAT) instructions on reaction and movement times. We used G*power[48] to estimate the necessary sample size for a significant effect of SAT instructions on these two parameters and found an effect size of $dz = 2.3$ for reaction times and 1.2 for movement times for a paired comparison. Given an alpha of 0.05 and power of 0.9 this resulted in a required sample size of $n = 10$ for the lower effect size (movement times).

Due to the invasive nature of STN recordings and DBS we were not able to record pilot data for computing the effect size of STN LFP changes or DBS effects. However, given the very good signal-to-noise ratio of invasive STN LFP recordings with a typical sample size of ~10[16,18,20] and the large effect size of previous studies testing DBS effects during perceptual decision-making (dz ~1.8[49] and ~2.4[40]) we considered the sample size estimation based on the behavioral measures appropriate. To also allow for possible drop-outs we opted to include 15 participants.

### Participants
Fifteen patients with PD, who had undergone STN DBS surgery prior to the experimental recordings, were recruited at two DBS centers; University Medical Center at the Johannes Gutenberg University Mainz, Germany ($n = 13$), and King's College Hospital London, UK ($n = 2$). See supplementary table 1 for clinical details. Lead localization was verified by monitoring the clinical effect and side effects during the operation, as well as through postoperative stereotactic computerized topography (CT), see supplementary Fig. 8. Recordings of bilateral STN LFP and DBS were performed through the externalized electrode extension cables and took place in the immediate postoperative period 1–3 days after electrode insertion, before implantation of the subcutaneous pulse generator. All patients were tested on their normal dopaminergic medication. Previous studies have shown that decision thresholds are not affected by dopaminergic medication in healthy people[50] or PD patients[51] and that medicated PD patients show similar adaptations of decision-making parameters during the SAT as healthy people[16]. To further confirm that patients' behavioral adjustments during the SAT were comparable to those of HC participants we enrolled 15 people without any neurological or psychiatric conditions. The two groups were matched for age (PD: mean 67.4 years, range: 49–79; HC: mean 67.5 years, range: 57–81; $P = 0.982$, independent samples $t$ test), handedness (1 left-handed person in each group as revealed by self-report, $P = 1$, Fischer's exact test) and gender (13 male in PD group, 10 male in HC group, $P = 0.39$, Fischer's exact test). In accordance with the declaration of Helsinki, all participants gave written informed consent to participate in the study, which was approved by the local ethics committees (State Medical Association of Rhineland-Palatinate and Oxfordshire REC A). One of the included PD patients (PD13) dropped out of the study because of post-operative fatigue. Another patient (PD09) was not included in the LFP analysis, because prior to this surgery he already had DBS electrodes implanted in the thalamus (ventral intermediate nucleus) for tremor-dominant PD, which evoked artifacts in the STN electrodes. In total, 13 patients performed the first session and 10 patients (see supplementary table 1) participated in the second session with STN burst stimulation (see below).

### Experimental task
We used a moving dots task probing the SAT, see Fig. 1A, B. Cues were presented on a MacBook Pro (MacOS Mojave, version 10.14.6, 13.3 inch display, 60 Hz refresh rate) using PsychoPy v1.8[52] implemented in Python 2. The display was viewed from a comfortable distance of

~50 cm. At the beginning of each trial a text cue indicated whether participants should respond as quickly (English: "Fast!"; German: "Schnell!") or as accurately as possible (English: "Accurate!"; German: "Akkurat!") for an average duration of 1 s (randomly jittered between 0.75 and 1.25 s) in randomized order. Then, a cloud with a diameter of 14 cm that consisted of 200 randomly moving white dots (dot size: 10 pixels) was shown on a black background. Each dot moved in a straight line at a rate of 0.14 mm per frame for 20 frames before moving to another part of the cloud where it moved in a new direction chosen pseudorandomly between +180 degrees and −180 degrees. Since 8 % of the dots moved coherently in one direction, while the remaining dots were moving randomly, the cloud of dots appeared to move to the left or right. Participants were instructed to respond with their right hand when and if they perceived that the cloud was moving to the right and with their left hand when and if they perceived a leftward movement. The trial was terminated by a response or after a fixed deadline, which was set at 3 s for accuracy trials and 2 s for speed trials. This was followed by immediate visual feedback, which was shown for 500 ms. During accuracy instructions "incorrect" (German: "falsch") was shown as feedback both for errors of commission and errors of omission, while "correct" (German: "richtig") was shown for all correct trials. During speed instructions "in time" (German: "rechtzeitig") was shown for all responses within the 2 s window, while "too slow" (German: "zu spaet") was shown if patients did not respond within the deadline. Responses were indicated by pressing a hand grip dynamometer (MIE Medical Research, Leeds, U.K.), which the participants held in each hand with their forearm comfortably positioned on the armrest of the chair. The analog force measurements were analog-to-digital converted and sent to the PsychoPy software through a labjack u3 system (Labjack Corporation, Lakewood, CO, USA) as well as to the LFP recording device (TMSi porti, see below). When the grip force exceeded a fixed threshold set at ~20 Newton a response was triggered. This threshold was perceived as relatively effortless and at the same time prevented triggering of involuntary responses, e.g., by tremor. When a response was triggered a TTL pulse was sent from Psychopy to the recording software through the labjack system so that task events were synchronized with the force and LFP recordings (as well as DBS bursts in the second experiment, see below). One session consisted of 200 trials and lasted ~10 minutes. All participants practiced the task for 40 trials before commencing the recordings.

### Analysis of behavioral data

All trials without responses (errors of omission), more than one response (i.e., if participants pressed the grippers twice in a trial), and response times <0.25 s were excluded[16]. For each trial the beginning of the force exertion was marked manually where a clear offset from baseline was visible at a fixed temporal resolution (1 s/inch) and blinded to trial type. We also applied an automatic movement onset detection algorithm where the onset was marked when the force crossed a pre-defined threshold. This threshold was defined as baseline (1 s to 0.5 s before the response was triggered) + 5* standard deviation of the baseline. Since the automatic detection led to incorrect onset detection in some trials, in particular in patients with action tremor (see supplementary Fig. 9), we used a manual definition of movement onset for this study. However, both methods led to similar movement time estimates (rho = 0.637, $P < 0.0001$, Pearson correlation). The following variables were computed: Reaction times (onset of moving dots cue until beginning of movement), movement times (beginning of movement until peak force), peak force (maximum force subtracted by baseline force) and response accuracy, see Fig. 1C, D.

Statistical analyses were conducted using hierarchical Bayesian regression models (linear mixed models) implemented in R-Stan (v2.21.2, https://mc-stan.org) using the rethinking (v2.13, https://github.com/rmcelreath/rethinking) and rstanarm (v2.21.1, https://mc-stan.org/rstanarm) packages. Single-trial values of reaction times,

movement times and peak force of each subject $j$ were log-transformed (because of their skewed distribution) and assumed to be drawn from a normal distribution with mean $\mu_j$ and standard deviation σ. Raw and log-transformed data are shown in supplementary Fig. 10 and results did not change when using non-log-transformed data (supplementary table 4). In each model we estimated the effect of Instruction (speed vs. accuracy), Group (PD vs. HC), and their interaction (IA) on the dependent variable:

$$\mu_j = \beta_{0j} + \beta_{1j}\text{*Instruction} + \beta_2\text{*Group} + \beta_3\text{*Instruction*Group} \quad (1)$$

All parameters were assumed to be drawn from normal distributions. To test whether Instruction affected reaction times depending on response accuracy we included this predictor along with the predictors from model (1) in an additional regression model.

We also conducted two control analyses to assess whether movement times were affected by response side or by fatigue throughout the experiment and whether this differed between groups (here shown for response side):

$$\mu_j = \beta_{0j} + \beta_{1j}\text{*Instruction} + \beta_2\text{*Group} + \beta_{3j}\text{*Response side} \\ + \beta_4\text{*Instruction*Group} + \beta_5\text{*Response side*Group} \quad (2)$$

Response accuracy at each trial was modeled as a binary distribution that a trial was correct (1) with probability $p$ and the effects of Instruction, Group, and their IA were estimated as follows:

$$\text{logit}(p) = \beta_{0j} + \beta_{1j}\text{*Instruction} + \beta_2\text{*Group} + \beta_3\text{*Instruction*Group} \quad (3)$$

where logit($p$) reflects the log-odds ratio that a trial is correct.

Finally, we assessed a putative relationship between reaction times and movement times at the single-trial level. We used reaction times as the dependent variable and included the z-scored single-trial movement times values along with Instruction and Group as predictors. Since we found it conceivable that the relationship between movement times and reaction times might depend both on Group and Instruction, we included all IA:

$$\mu_j = \beta_{0j} + \beta_{1j}\text{*Instruction} + \beta_2\text{*Group} \\ + \beta_{3j}\text{*movement times} + \beta_4\text{*Instruction*Group} \\ + \beta_5\text{*movement times*Group} + \beta_6\text{*movement times*Instruction} \\ + \beta_7\text{*movement times*Instruction*Group}$$

$$(4)$$

We used non-informative priors in all models. All scripts including the priors are available on https://data.mrc.ox.ac.uk [53]. For each model, 3 Markov Chain Monte Carlo (MCMC) chains were run using 5000 iterations each of which the first 500 were discarded as burn-in. Convergence was assessed using the Gelman-Ruben r-hat statistic. Statistical inference was based on the 95% Bayesian Credible Interval (CrI) of the posterior distribution, i.e., the highest density interval containing 95% of the distribution. If the 95% CrI did not include 0, the effect was interpreted as significant. For illustrations of the posteriors the bayesplot (v1.8.0, https://mc-stan.org/bayesplot) and ggplot2 (v3.4.0, https://ggplot2.tidyverse.org) packages were used. CrI of the (non-back transformed) posteriors are reported in the text when appropriate and CrI of all statistical tests are listed in supplementary table 2.

We also conducted the frequentist equivalents of these analyses with non-Bayesian linear mixed effect models and 95% confidence intervals (CI) using lme4 (v1.1.30)[54] implemented in R (v4.0.5) and report all results in supplementary table 2 along with $t$- and $p$ values. None of the results differed regarding their significance depending on whether the Bayesian or frequentist method was used demonstrating that the results were not affected by sensitivity to the choice of priors.

Any statistical tests that did not involve single-trial data were performed using frequentist $t$ tests for group comparisons of continuous data, Fisher's exact test for comparisons of binary data (e.g., handedness), and linear regression for regression analyses across participants. Post hoc tests were corrected for multiple comparisons using the Bonferroni method and corresponding P-values are marked as $P_{corrected}$. Two-tailed P-values were used throughout the statistical analyses except when there were clear hypotheses about the directionality of effects, in which case one-tailed values were used (marked as $P_{one-tailed}$). Notably, this was only the case for the post hoc comparison between contralateral and ipsilateral DBS (see results).

## Drift diffusion modeling

DDM models the decision-making process as a continuous integration of sensory evidence for two alternative options until a threshold is reached, which reflects that sufficient evidence has been accumulated to commit to a choice[26]. There are three main parameters: the drift rate $v$ reflects the rate of evidence accumulation (with low drift rates resulting in slow and error-prone choices), the decision threshold $a$ defines the amount of evidence that is needed before responding (with low thresholds resulting in faster but less accurate responses), and the non-decision time $t$ reflects processes not directly related to the decision process, such as afferent delay, early sensory processing, and movement preparation, see Fig. 2A. Of note, since we here predicted reaction times that did not contain movement (see above), the non-decision time parameter did not include movement execution. In a control analysis, we included an additional bias parameter defining whether the accumulation process started centered between the two options or might be biased toward one of them. This parameter was not different from 0.5 (i.e., centered) according to its 95% CrI in the initial model (see below) and did not alter any of the reported results when including it in the models. Therefore results from the less complex models without this parameter are reported in the main text. We applied a Bayesian hierarchical estimation of DDM (HDDM v0.8.0)[27] implemented in Python3 (v3.6). Analogously to our behavioral analysis, the hierarchical design assumes that parameters from individual participants are not completely independent, but drawn from a common group distribution. Prior distributions were informed by 23 previous studies[27]. While we hypothesized that speed vs. accuracy instruction would primarily modulate the decision threshold parameter[14,16,26], we initially also allowed drift rate and non-decision time to vary depending on instruction and estimated this full model both for PD patients and healthy people. MCMC sampling was used for Bayesian approximation of the posterior distribution of model parameters generating 10,000 samples and discarding 2000 samples as burn-in. Convergence was assessed by inspecting traces of model parameters, their autocorrelation, and computing the Gelman-Rubin R-hat statistic[27]. Parameters of the model were analyzed for significance by computing the CrI of the posteriors and tested for significance analogously to the behavioral analysis described above. When we had clear a-priori hypotheses concerning the directionality of effects (since HDDM was conducted after the behavioral analysis), we also considered one-tailed tests, i.e., the 90% CrI testing whether the distribution overlapped with 0 on one side of the distribution, significant. In these cases, this is clearly marked as $CrI_{90}$ when reporting the results.

After estimating this full model we then reduced the model to only contain the significant modulations (i.e., effect of Instruction on decision thresholds, see results) and assessed model performance using quantile probability plots[26], in which predicted and observed reaction times for the 10, 30, 50, 70, and 90 percentile were plotted against their predicted and observed cumulative probability for each condition[16,20]. To assess whether single-trial changes in ($z$ scored) movement times were related to changes in decision-making

parameters, we then applied HDDM regression analysis using the following model:

$$a = \beta_{0j} + \beta_1 * \text{Instruction} + \beta_2 * \text{movement time} + \beta_3 * \text{Instruction} * \text{movement time}$$
(5)

$$v = \beta_{0j} + \beta_1 * \text{movement time}$$
(6)

$$t = \beta_{0j} + \beta_1 * \text{movement time}$$
(7)

We estimated this for PD patients and healthy people. Statistical analyses were performed as described above.

## Processing of STN LFPs

LFPs were sampled from bilateral STN at 2048 Hz, bandpass filtered between 0.5 and 500 Hz and amplified with a TMSi porti device (TMS International, Enschede, The Netherlands). The same system was used for recording the force measures and TTL pulses (see above) through auxiliary input channels. The whole recording was visually inspected for artifacts off-line in Spike2 (Cambridge Electronic Design, Cambridge, UK) and noisy trials were rejected. After artifact rejection (on behavioral and neurophysiological grounds) ~140 trials per patient and 1780 trials in total remained. Further analysis of the data was performed using FieldTrip (v20201126)[55] implemented in Matlab (R 2019a, The MathWorks, Natick, MA, USA). All scripts are available on https://data.mrc.ox.ac.uk[53]. The data were imported to Matlab, high-pass filtered at 1 Hz using a 4th order Butterworth filter, bandstop filtered between 49-51 Hz (FieldTrip function *ft_preprocessing*), and downsampled to 200 Hz using an anti-aliasing filter at 100 Hz (*ft_resample*). A bipolar montage was created from the monopolar recordings by computing the difference between the most dorsal omnidirectional contact and the neighboring three dorsal directional contacts, between the three dorsal and corresponding three ventral directional contacts, as well as the three ventral directional contacts and neighboring most ventral omnidirectional contact resulting in 9 bipolar channels per STN (*ft_apply_montage*). In one patient (PD2), who was implanted with quadripolar (i.e., non-directional) leads, three bipolar contacts per STN were created by computing the difference between the neighboring contacts (according to Medtronic™ contacts 0 vs. 1, 1 vs. 2 and 2 vs. 3). For each bipolar channel the data were transformed to the frequency domain using the continuous Morlet wavelet transform (width = 7, *ft_freqanalysis*) for frequencies from 2 to 100 Hz using steps of 1 Hz and 20 ms throughout the whole recording. Power of each frequency was baseline corrected (*ft_freqbaseline*) relative to the mean power of that frequency across the whole recording[16]. The resulting spectra were epoched and aligned with, respectively, onset of the moving dots cue and movement onset (*ft_redefinetrial*). Spectra of each single-trial were classified as contralateral and ipsilateral with respect to the effector (left or right hand) of the current trial. In order to identify the bipolar contact, which showed the strongest task-related modulation, we analyzed all contacts with respect to their expression of movement-related gamma (55–80 Hz) activity. We chose this frequency as a functional localizer, because STN gamma activity is highly focal (in contrast to lower frequencies), localized within the dorsal STN[17,35] and correlates with movement parameters during force production[17,18]. For each hemisphere the contact with the strongest movement-related increase in gamma was chosen for further analysis. If no clear gamma increase could be detected (8/26 sides), the channel with the strongest beta power (defined individually between 13 and 30 Hz) reduction during the movement was chosen for each STN. We confirmed the validity of this functional localizer approach by conducting lead localization analysis (see below). For each trial activity was extracted from the STN contralateral to the movement resulting in one STN channel per patient.

### Electrode localization

Electrode localization was carried out using the Lead-DBS toolbox (v.2.5.2; https://www.lead-dbs.org/) with default parameters as described elsewhere[56]. Briefly, using Advanced Normalization Tools (ANTs) preoperative magnetic resonance imaging and postoperative CT scans were corrected for low-frequency intensity non-uniformity with the N4Bias-Field-Correction algorithm, co-registered using a linear transform and normalized into Montreal Neurological Institute (MNI) space (2009b non-linear asymmetric). Brain shifts in postoperative acquisitions were corrected by applying the "subcortical refine" setting as implemented in Lead-DBS[57]. The reconstructed electrodes (marked at contacts, which were used for LFP recordings and stimulation) were then overlaid on the STN to confirm proper targeting, see supplementary Fig. 8. Imaging data was not available in three patients. In these patients we only relied on the functional localizer, but proper placement was also suggested by the clinical improvement during continuous DBS (see supplementary table 1), which would not be expected with placement outside of the STN.

### Statistical analysis of STN LFPs

For the statistical analysis of LFPs we had clear a-priori hypotheses about the temporal and spectral characteristics of STN activity relevant for adjustments of, respectively, decision thresholds and movement parameters based on previous studies.

(i) Movement-related increase in gamma ($Gamma_{move}$) activity and

(ii) movement-related decrease in beta ($Beta_{move}$) activity, both of which have been shown to correlate with movement parameters during force production[17,18].

(iii) Cue-related decrease in beta activity ($Beta_{cue}$), which has been shown to be modulated by speed vs. accuracy instructions, and to correlate with changes in reaction times and decision thresholds[16,40,58].

(iv) Cue-related changes in theta ($Theta_{cue}$: 4–8 Hz) activity, which have been related to adjustments of reaction times[28] and decision thresholds[16,20].

We analyzed whether changes in STN activity were related to trial-by-trial adjustments of reaction and movement times after extracting single-trial values as the mean value in time windows, which were based on the features in spectra averaged across all trials (i.e., irrespective of speed vs accuracy). These features and respective time windows were:

(i) $Gamma_{move}$ from movement onset to 300 ms after movement onset (see supplementary Fig. 3B showing the clear, temporally confined gamma increase).

(ii) $Beta_{move}$ using the same time window as above, see Fig. 3D.

(iii) $Beta_{cue}$ reflecting the reduction of beta power after the cue[16,40], i.e., the difference between beta power at stable pre-cue levels (300-100 ms precue) and after the decrease where beta power again plateaued (320-400 ms post cue; the trough was at 320 ms), see Fig. 3B. The plateau at ~400 ms has consistently been observed across studies[16,40]. To ensure that no movements occurred during this time window (beta power strongly decreases during movement) we excluded all trials with reaction times <400 ms (7% of all trials) for the prediction of reaction times (see below) as a control analysis. While the post cue window (320-400 ms) could not be moved to later time periods to avoid increasing numbers of responses occurring in this window, we conducted control analyses shifting the baseline period across different time windows of the pre-cue period demonstrating that the results were not dependent on the exact definition of the pre-cue window (supplementary table 5).

(iv) $Theta_{cue}$ extracted from a 750 ms time window after the cue based on our previous study[16]. To ensure that no movements occurred during this time window we excluded all trials (30% of all trials)

with reaction times <750 ms for the prediction of reaction times (see below).

All single-trial values were z scored by subtracting the mean and dividing by the standard deviation for each patient. Trials with z scores >3 were excluded (<5% of trials combined).

We used the same Bayesian hierarchical regression models as described for the behavioral analysis, now including $Gamma_{move}$, $Beta_{move}$ and $Beta_{cue}$ as predictors:

$$\mu_j = \beta_{0j} + \beta_1*Instruction + \beta_2*Gamma_{move} + \beta_3*Beta_{move} + \beta_4*Beta_{cue}$$
$$+ \beta_5*Gamma_{move}*Instruction + \beta_6*Beta_{move}*Instruction + \beta_7*Beta_{cue}*Instruction$$

(8)

This was done using reaction and movement time as dependent variable in two separate models thus allowing us to assess whether single-trial changes in the respective frequency bands were predictive of the behavioral adjustments. Since for $Theta_{cue}$, a larger amount of trials had to be excluded (~30% had reaction times <750 ms) we performed this analysis in a separate regression model only containing $Theta_{cue}$, Instruction and its interaction. Since only beta power was related to both adjustments of movement and reaction times, results from theta and gamma power are mainly reported in supplementary Fig. 3. Finally, we assessed a putative by-trial relationship between $Beta_{cue}$ and $Beta_{move}$ by using the former as a predictor of the latter.

We also assessed whether changes in STN activity in the described time windows were modulated by task instructions. To this end power of each frequency during the respective time windows was averaged across trials separately for speed and accuracy trials and this trial-averaged data then compared using paired $t$ tests.

### HDDM with single-trial STN LFPs

After having established which of the frequency bands were related to adjustments of reaction times, we analyzed whether they were specifically related to changes in distinct decision-making parameters. To this end, we entered the respective single-trial values into an HDDM regression model. Due to the different amount of trials and to reduce model complexity, this was done for each frequency band separately and only if the respective frequency bands showed significant effects on reaction times in the previous analysis (here shown for $Beta_{cue}$):

$$a = \beta_{0j} + \beta_1*Instruction + \beta_2*Beta_{cue} + \beta_3*Instruction*Beta_{cue} \quad (9)$$

$$v = \beta_{0j} + \beta_1*Beta_{cue} \quad (10)$$

$$t = \beta_{0j} + \beta_1*Beta_{cue} \quad (11)$$

Sampling and statistical analysis of HDDM were performed as described above.

### Burst stimulation

After the first session patients had a short break of ~30–60 min. During this time the LFPs recorded from bilateral STN were processed and analyzed as described above, but instead of constructing bipolar channels from neighboring electrodes, two wider bipolar contacts were constructed so as to allow recording during stimulation of an intervening contact. First, directional contacts were averaged to form an omnidirectional contact (resulting in four omnidirectional contacts per STN). Then, a dorsal bipolar contact between the most dorsal and second most ventral contact and a ventral bipolar contact between the most ventral and the second most dorsal contact were created. This was done to compute the bipolar contact with the clearest movement modulation of gamma and beta activity, since this has been related to localization within or close to the dorsal STN[17,24,34,35], and allows

stimulation of the contact in between this bipolar pair to mitigate the stimulation artifact using common-mode rejection[40,59]. The two bipolar contacts on each side were then compared regarding the extent of movement-related gamma and beta modulation and the best contacts (i.e., with the clearest modulation) chosen as recording electrodes using the electrode in between as active contact for stimulation. DBS was applied using a custom-built device previously validated[40] in pseudo-monopolar mode using reference pads on the patients' shoulders as anode. Frequency (130 Hz) and pulse width (60 µs) were fixed. To allow inference on timing-specific effects of stimulation DBS was applied in bursts. Mean DBS burst duration was 250 ms (drawn randomly from a uniform distribution between 150 and 350 ms) and mean burst interval was 150 ms (drawn randomly from a uniform distribution between 75 and 225 ms), see Fig. 4A. These parameters were defined based on our previous study of closed-loop DBS[40] and were in simulations shown to result in DBS bursts occurring in ~50% of trials in any given 100 ms time window during the experimental task allowing us to compare timing-specific behavioral effects of DBS vs. no-DBS. Stimulation was applied simultaneously to both hemispheres and ramped up and down to reduce paresthesia[40,59]. Ramp duration depended on the DBS intensity and ranged from ~150 to ~350 ms (see supplementary table 1 for details). DBS intensity was titrated by slowly increasing the intensity of continuous DBS on each side and evaluating clinical effects on Parkinsonian symptoms as well as putative side effects by a trained clinician. When the threshold for clinical effects was reached the intensity was noted and, in the case of side effects, slightly decreased. We evaluated this procedure by performing double-blind UPDRS-III scores (limb bradykinesia, rigidity, and tremor scores) in continuous DBS ON vs. OFF. This showed an improvement in clinical scores in each patient (average from 25.3 to 17.8, $P < 0.001$, paired samples $t$ test) confirming that the chosen intensities were clinically effective. We then used this intensity for burst stimulation while patients performed the same experimental task as described above. None of the patients reported paresthesia during the experiment.

In a third and final session, PD patients again performed the same experiment with STN burst stimulation but instead of bilateral stimulation, DBS was given unilaterally. Respectively left and right STN were simulated in separate sessions comprising 100 trials each in counterbalanced order. Other than that simulation settings were identical. Two of the patients who underwent bilateral stimulation were not able to perform the final session due to fatigue leaving eight patients for this session.

## Effects of burst stimulation on behavior

Timing-specific effects of STN burst stimulation were analyzed using a moving-window approach[40]. Stimulation intensity at each sample was saved in the recording software and imported to Matlab along with the TTL pulse (signaling the response), downsampled to 200 Hz, and binarized (0 for no stimulation, 1 for stimulation). Since intensities during ramping up and down of stimulation were below the clinically effective intensity they were defined as no stimulation[40]. However, since stimulation might already have effects on decision-making processes at intensities below clinical thresholds[43] we conducted additional control analyses defining incremental parts of ramping as stimulation. This was done until inclusion of 100 ms ramping (the last 50 ms of ramping up and first 50 ms of ramping down), since here only 20% of trials remained as off-stimulation trials.

For each trial, we noted for 100 ms long time windows if stimulation was applied or not (at any point during that window). This time window was shifted by 10 ms over 1500 ms (from −500 to +1000 ms for the cue-aligned data and from −1000 to +500 ms for the movement-aligned data). We also analyzed the percentage of trials in which stimulation was applied at any given time window, which confirmed that stimulation was applied between ~40 and 50% of trials at all

time windows (Fig. 4B). Next, for each time window, we computed the average reaction and movement time after speed vs. accuracy instructions for all trials in which stimulation was applied and all trials in which stimulation was not applied. At the second level, i.e., in the across-subjects analysis, we then compared whether the SAT effect ($RT_{Accuracy}−RT_{Speed}$) was affected by stimulation by performing cluster-based permutation tests[40,60,61]. At each time window, we computed the effect of stimulation on the SAT using an alpha of 0.05 as cluster-building threshold. To correct for the high number of statistical tests the resulting clusters, which consisted of all time points that exceeded the initial threshold, were compared against the probability of clusters occurring by chance by randomly shuffling between stimulation labels (stimulation versus no stimulation) of each subject using 1000 permutations. Only clusters in the observed data that were larger than 95% of the distribution of clusters obtained in the permutation analysis were considered significant and marked as $P_{cluster} < 0.05$. In case of a significant effect, post hoc tests were conducted by extracting the mean values of the respective time windows (termed $DBS_{RT}$ and $DBS_{MT}$) for each patient and then tested for stimulation effects during speed and accuracy trials using one-sample $t$ tests. We also assessed whether stimulation in the time window affecting reaction times ($DBS_{RT}$) modulated decision thresholds (as we hypothesized from the previous analyses, see Results) by using HDDM including DBS (stimulation vs. no stimulation) as a predictor of thresholds during respective speed and accuracy trials. Due to the limited number of trials (10 patients, and given that DBS at each time window was only applied in ~50% of trials) we did not use more complex models.

For unilateral burst stimulation, processing of the data was identical to bilateral stimulation except that trials were divided into responses with the contralateral vs. ipsilateral hand, since we here asked whether the observed effects of stimulation on reaction times were global (i.e., related to withholding any response) or lateralized (i.e., responses of the contralateral hand). To assure a sufficient number of trials, they were initially not further subdivided into speed vs. accuracy, but we compared the conditions in a post hoc test for the significant time window (see results). Based on the results from the previous bilateral stimulation analysis, we computed the reaction times for trials with contralateral and ipsilateral responses for the time windows aligned to the cue and computed whether they were affected by stimulation using permutation tests, shuffling the stimulation labels (see above). To directly compare effects of contralateral and ipsilateral stimulation on reaction times we extracted the mean values of the significant time window (termed $DBS_{contra}$, see results) for each patient and then compared the two stimulation conditions using a paired $t$ test. We also included DBS at this time window as a predictor of decision thresholds in HDDM for contralateral and ipsilateral stimulation. Finally, to assess whether the behavioral effects of unilateral stimulation were related to the significant effects on reaction time adjustments that we had observed during bilateral stimulation, we performed linear regression using behavioral effects of unilateral stimulation as predictor and of bilateral stimulation as a dependent variable for, respectively, contralateral and ipsilateral stimulation. To directly compare contralateral and ipsilateral stimulation we also included both as predictors in a multiple regression analysis.

## Effects of burst stimulation on STN LFPs

Whilst stimulation was applied LFPs were continuously recorded through the two contacts neighboring the stimulation contact and a bipolar signal derived as previously described (i.e., wide bipolar recording). Despite common-mode rejection, the artifact was clearly visible (see supplementary Fig. 5A) and its spectral characteristics were not strictly confined to the stimulation frequency and its harmonics (supplementary Fig. 5B, C). Hence the following artifact removal procedure was applied. The data were imported to Matlab, high-pass filtered at 4 Hz and low-pass filtered at 100 Hz using a 4th order

butterworth filter, demeaned, and detrended (*ft_preprocessing*). After visual inspection of the LFPs from each patient a common threshold was set at 10 μV. This was chosen, because the remaining (i.e., after filtering) stimulation artifact, but not physiological LFPs (in the interval of stimulation bursts), consistently crossed this threshold. At each sample the signal was removed if it crossed the threshold (on average <1% of the signal) and replaced by linear interpolation of the neighboring non-noisy signals. Afterwards the data were downsampled to 200 Hz and the subsequent time-frequency analysis was identical to the LFP recordings described above. For each patient at least one hemisphere showed clear beta power reduction around the movement, and the best hemisphere was then chosen for the second-level analysis (an example of single subject beta power is shown in supplementary Fig. 5D, E, subject-averaged spectra are shown in supplementary Fig. 6). We also assessed the overall effect of stimulation on beta power by aligning beta power to the onset of stimulation (after ramping) and normalizing it to the mean beta power when no stimulation was applied (i.e., during the stimulation interval). As expected[29–32], this showed a clear stimulation-related reduction in beta power, see supplementary Fig. 7.

To assess the electrophysiological effects of timing-specific stimulation (i.e., in the time windows affecting behavior), on the second level we computed the cue-related beta power change for DBS$_{RT}$ and DBS$_{MT}$ trials and compared them to trials where no stimulation was applied in the respective windows (500 ms windows centered around the respective DBS windows, i.e., −300 to +200 ms for DBS$_{RT}$ and +200 to +700 ms for DBS$_{MT}$) using cluster-based permutation tests as described above. Of note, since beta power was normalized to the mean power across the whole stimulation session stimulation-induced reductions in beta power would result in relatively higher beta power when no stimulation was applied. To ensure that changes were not due to responses occurring earlier when stimulation was applied (the DBS$_{MT}$ window occurred relatively late during the trial), we only assessed beta power at each trial until onset of the movement. In more detail, for a 1 s long window (400 ms before the cue to 700 ms after the cue) the time series of each trial was capped at the time of movement onset and then these time series were averaged across trials[16]. To ensure a sufficient number of trials we did not further subdivide trials into speed vs. accuracy.

For unilateral stimulation, the same processing and analysis of LFP data were carried out as described above, except that we compared the stimulated STN (for trials with stimulation at the critical time window) to the STN that was not stimulated (since stimulation was only applied in one hemisphere at a time) thus allowing us to assess whether the observed changes were due to stimulation or simply due to differences in data acquisition (neighboring vs wide bipolar recordings).

Of note, we did not attempt to analyze gamma or theta power during stimulation since these frequency bands were heavily affected by stimulation artifacts (see Supplementary Figs. 5 and 6).

### Reporting summary
Further information on research design is available in the Nature Portfolio Reporting Summary linked to this article.

## Data availability
Original data are available upon request to the corresponding author (damian.m.herz@gmail.com). At present, participant consent does not allow for depositing the full original dataset. A minimum example dataset (including scripts) is available on https://data.mrc.ox.ac.uk/data-set/subthalamic-nucleus-correlates-decision-and-movement-speed (https://doi.org/10.5287/bodleian:1R9KzGXxM). Source data are provided with this paper.

## Code availability
Code (including instructions and a minimum example dataset) is available on https://data.mrc.ox.ac.uk/data-set/subthalamic-nucleus-correlates-decision-and-movement-speed (https://doi.org/10.5287/bodleian:1R9KzGXxM).

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

## Acknowledgements

D.M.H. is supported by a postdoctoral grant from the Independent Research Fund Denmark (0168-00014B). H.T. and P.B. are supported by the Medical Research Council (MC_UU_00003/2). R.B. is supported by the Medical Research Council (MC_UU_00003/1).

## Author contributions

Conception and design: D.M.H., R.B., S.G., P.B. Acquisition, analysis, and interpretation of data: D.M.H., M.B., G.G.E., M.A., K.A., P.F., H.T., R.B., M.M., S.G., P.B. First draft of manuscript: D.M.H. Revision of manuscript: M.B., G.G.E., M.A., K.A., P.F., H.T., R.B., M.M., S.G., P.B.

## Competing interests

The authors declare no competing interests.
