## [Peer Review File · Nature Communications]

Dynamic control of decision and movement speed in the human basal gangliaREVIEWER COMMENTS

Reviewer #1 (Remarks to the Author):

The authors speculate that the role of STN in decision-making is to control the release of a motor response. And to disentangle the cognitive decision-making and movement control, they designed the bilateral and unilateral DBS stimulation paradigm. The hypothesis is validated in a causal manner through the concrete evidence that only contralateral hand reaction time was affected. The manuscript is well-written and completes fancy computational modeling and data analyses. Especially, the artifact correction methods used for recovery of beta power modulation during stimulation trials are elegant.

Though it is honest to say that this study is well done and provides new knowledge, I hold major concerns regarding the design methods and interpretation of results. These major concerns, together with some minor ones, are listed below.

Major concerns on conceptual design of this study.

1. Based on previous correlative evidence 10-12, the authorship hypothesized that the correlates and causal contributions of STN to deliberation precedes its effect on movement speed. A major issue here is that the too roughly-named 'deliberation' must be a complex structure containing many processes which may influence both decision accuracy and reaction time/speed. It seems that the authors overweighted speed but relatively devalue the accuracy. Their distinction should be explicitly informed and the focus of the whole study should be clearly stated.

2. Related to the above issue of accuracy, a critical finding is that though PD patients had lower accuracy rates compared to healthy people accuracy rates were not significantly different between Speed and Accuracy instructions. This at least partially contradicts the main goal of the study which aims to evaluate to what extent decision and movement processes are controlled by common or separate mechanisms. Since accuracy rate is a direct and critical measure for decision-making, the authors may have mainly measured movement-related processes but largely failed to measure decision-making processes (e.g., general cautiousness). Unfortunately, this pitfall in paradigm design may have rendered the global effect, if any, just undetectable.

3. Previous evidence linking STN to the computation of 'higher-order' decision-making, especially to the agent's general level of cautiousness, should be described in more detail. For example, to what extent a higher-order decision structure must exist globally? Is it possible that a from low to high hierarchy indeed exists but independently works for each hemisphere? These issues should be clarified and discussed more thoroughly.

4. In addition, I note that this study is similar to the Current Biology. 2018 paper regarding the paradigm used, the analyzing pipeline, and some of the results reported. The authors may elaborate on the common points and differences between the two studies, otherwise the novelty of this paper can be considerably reduced.

Major concerns on interpretation for results.

1. An interesting and important finding is that 'at the single trial level longer reaction times were related to faster movements and this relationship did not differ depending on group or instructions'. I wonder what psychological mechanism underlies this negative correlation. One possibility is that longer reaction time affords a better preparation for the subsequent movement, providing stronger forces and thus faster speeds. However, this possibility is challenged by findings in beta power activity, which show that the beta band activities, though temporally separates decision and movement time in different window, are statistically independent from each other. Currently I do not see a clear path linking above findings together.

2. In line 278 and Figure 5B, the authors stated that 'during DBSRT compared to DBSMT beta power was reduced from 20 ms precue to 160 ms postcue. Several points are unclear. First, the ratio of trials with DBS stimulation during both RT and MT is unshown. It is unclear to what extent DBSRT /DBSMT contains stimulation during RT/MT phase only. Second, it is a little puzzling to me why the effect of DBSRT appears mainly at postcue period (0-160ms). What is the interpretation? Third, in the middle panel of figure 5A stimulation windows have been highlighted for both RT and MT. As a direct validation, it is necessary to additionally show results for trials with stimulation in only these two windows.

Major concerns on methodological issues.

1. The authors included both correct and incorrect trials into analysis. If results here are true, they should remain if only correct trials are left and incorrect trials removed. Though this will sacrifice 25%~35% incorrect trials (reference to 65%~75% accuracy rate reported in line 94), I suggest the authors rerun the analysis following this strategy. Theoretically, incorrect trials are more susceptible to noise and this further explains why incorrect trials should be excluded.

2. In terms of stimulation, I have several major concerns.

The authors used an approach of dynamic stimulation analysis to compare behavior and LFP differences between ON-stimulation and OFF-stimulation states. By setting parameters of burst stimulation (mean duration 250 ms, mean interval 150 ms), at any given 100 ms time-window, stimulation was applied to ~50% of trials. A ramping strategy was used to prevent sudden increases or decreases of the stimulation voltage.

First, time points during ramping were defined as no-stimulation. I do not believe this is the correct way to do. The ramping procedure is slow, lasting 130-350 ms. They stated that during this period, stimulation was below the effective intensity. This may be correct for motor improvement (not even so for time points when stimulation intensity is reaching the plateau), but not necessarily for cognitive processing. Therefore, I would at least divide some part of the ramping time to the ON-stimulation group (though the appropriate proportion is unclear). Or, more strictly, discarding data points during ramping since they may “contaminate” both the ON-stimulation and OFF-stimulation periods.

Second, the stimulation interval lasted on average 150 ms was defined as NO-stimulation. I doubt its rationality. This period better represents post-stimulation state rather than no-stimulation. It is clear that basal ganglia neurons cannot return to the pre-stimulation/no-stimulation state within a short 150 ms interval (instead, at least 2-3 s may need (see Science. 2021 paper)). Following this conception, all results based on these comparisons are actually revealing differences between with-stimulation and shortly-after stimulation (when neurons are still affected by the stimulation).

Third, DBS-induced changes in accuracy rate should be reported for both bilateral and unilateral stimulations.

Fourth, I recommend a comparison between activity from the contralateral vs. ipsilateral STN contacts, to evaluate to what extent signals are intrinsically isolated or shared between hemispheres.

Fifth, different with bilateral stimulation analysis (Figure 4D and 4F) the authors did not subdivide trials into Speed vs. Accuracy for unilateral stimulation (as explained by authors, in order to assure a sufficient number of trials). Though the authors explained that this was to assure a sufficient number of trials (line 803), I believe additional analyses are required to evaluate the behavioral modulation effect of unilateral stimulation for accuracy and speed trials, respectively. This will help to tell whether the effect of Instruction (Speed vs. Accuracy) is dependent on bilateral stimulation or not.

3.As stated in line 493, for each trial the beginning of the force exertion was marked manually where a clear offset from baseline was visible at a fixed temporal resolution (1 s/inch) and blinded to trial type. I wonder why the beginning point has to be marked manually. If possible, I recommend they determine these starting points using algorithms. This would make it more replicable for other researchers to follow.

4.In Figure 3B, the authors calculated beta power decrease as the differential mean power between window 300-100 ms precue and window 320-400ms postcue. How these two windows are determined?

To avoid p-hacking, a more rigorous and better solution is to let data say by itself. Similarly, why the window for Theta power analysis in Figure 4C switched to 0-700ms postcue?

5. In most cases, the authors only reported C_{rl} to evaluate the statistical significance. To help readers better evaluate the results, I recommend authors show more comprehensive measures, including but may be not restricted to correlation coefficient, regression coefficient degree of freedom and p-values, when necessary for their key findings.

Minor issues

1. In Processing of STN LFPs, Methods, according to the authors, for directional leads, bipolar rereference was made between “the most dorsal omnidirectional contact and the neighbouring three dorsal directional contacts, between the three dorsal and corresponding three ventral directional contacts, as well as the three ventral directional contacts and neighbouring most ventral omnidirectional contact”. Following this conception, 9 rather than 8 bipolar channels would be produced. Please check this.

2. In Processing of STN LFPs, Methods, the authors confirmed functional localizer approach by further conducting lead localization analysis. However, imaging data were not available in three subjects. Do the authors rely solely on the functional localizer results (which is also accepted), or do they conduct other analyses on these three subjects to confirm that the lead is within the dorsal STN? Please report.

3. In line 99, Accuracy rates...but participants committed errors disproportionately faster after Speed vs. Accuracy instruction as indicated by a significant Instruction*accuracy IA on reaction times (C_{rl} [+0.042:+0.223])... I wonder how to interpret the result and what psychological mechanism underlies this positive correlation. In addition, is it necessary to study the effect of the accuracy on the movement time as well?

4. In line 501, Single trial values of reaction times, movement times and peak force of each 502 subject j were log-transformed (because of their skewed distribution). I think it's better to plot the distribution of reaction times, movement times and peak force, and illustrate the parameters of the skewed distributions.

5. typo errors exist. For example, in line 281 figure 5B was mistakenly spelled as figure 5F. NOTE that several more similar errors have been noticed but not listed here.

6. To keep the whole study concise, I suggest placing theta and gamma-related results in the supplementary material.

Reviewer #2 (Remarks to the Author):

This incredibly interesting manuscript describes a rare set of behavioral, electrophysiological, and stimulation experiments in humans, focused on the subthalamic nucleus' (STN's) role in regulating decision and motor states. The authors had participants perform a simple variant of the dot motion task, indicating the left/right motion direction of the dots by squeezing a dynamometer in either hand. Participants were told via visually displayed cues whether to focus on speed or accuracy of their responses, which were then modeled using the drift diffusion model (DDM). Patients with deep brain stimulation electrodes implanted in the STN were tested alongside non-patient controls.

The authors first confirm prior reports that speed constraints on decision time relate to a compression of the boundary height from the DDM, which also appears to correlate with movement times. Next, they show that beta frequencies in the STN drop after the imperative cue and rebound after the response has been made, indicating a role in the decision and movement. Then, and most importantly, they found that there were two critical windows for stimulation that impacted task performance: 1) an early (10-180ms) period before the cue that decreased reaction times, and 2) a late (330-460ms) period that impacted movement times. The authors then show how stimulation in each of these windows disrupted STN beta responses. Finally, the authors show how the effects of stimulation on behavior and STN beta power are lateralized to the side of the brain contralateral to the responding hand. The authors conclude from this evidence that the STN is more likely to be involved in implementation of motor actions (specifically by withholding the prepared action), rather than decision-making.

All in all, I thought this was a very unique and interesting study that can contribute a lot to our understanding of the role of basal ganglia nuclei in decision making and motor implementation in humans. However, there are some inferential and narrative problems that make the power of the findings hard to see as a reader.

MAJOR CONCERNS

1. Inference.

If I understand this correctly, the authors conclude that the STN's role in this task is more motoric (i.e., withholding a planned action) than cognitive (i.e., computing the decision threshold) based primarily on 3 bits of evidence.

- Beta power in the STN did not appear to be modulated by speed-vs-accuracy cue instructions (though other frequency spectrums in the STN do).
- Stimulation decreased reaction times in the pre-cue window.
- Stimulation decreased movement times in the post-cue window.

I have three problems with this. First, the contrastive framing of the two alternative hypotheses (computing decision threshold vs. gating the motor response) is itself a sort of strawman of the competing theories on STN/basal ganglia literature on their roles during response selection. While many have pointed out a role for the STN (or broader indirect pathway) in modulating the decision threshold, many computational theorists posit that the threshold itself is either implemented downstream or as a distributed computation across multiple regions. On the other hand, the movement vigor hypothesis (a variant of the implementation hypothesis presented in this paper), states that the movement speed is regulated by a competition between the direct and indirect pathways (a similar mechanism has been postulated for modulating the drift-rate). Both of these more nuanced ways of understanding the computation of the STN make this experiment a less severe test (to borrow Popper's term) of each hypothesis.

Second, and perhaps more importantly, even within the narrow framing of the contrastive hypothesis that the authors adopted here, I am not sure that the evidence itself is strong enough to warrant the authors conclusion. Yes, reaction times appear to speed up when STN stimulation happens before the cue. But I don't see any analysis on the accuracy of responses. If response accuracy is also impacted by pre-cue stimulation, then this indicates a disruption of the decision process itself, rather than the motor response. Without this analysis it is hard to fully understand what is happening as a result of STN stimulation.

Finally, and related to the previous point, it is not clear why both hypotheses cannot be true. STN stimulation clearly had two windows of efficacy separated by several hundred milliseconds. Within the framing of the hypotheses presented by the authors, it seems possible that two phases of signal come through the STN: an early phase related to the decision and a later phase related to the execution. I am not certain how the current results cannot rule out this third hypothesis. Why was accuracy excluded as a primary outcome measure of interest, particularly when it's relevant to the DDM?

2. Analysis

I really liked the careful DDM analysis used in this paper. However, I am a bit confused as to why the bias term (z) was not included as a free parameter in the early model evaluations. This seems like an incomplete vetting of the effects of task (and stimulation) on the underlying decision parameters. While I believe the results shown here focused on the decision threshold, particularly since they align with many previous reports, it is still possible that including bias (z) could change some of the results shown here.

3. Vigor

There are two ways that the basal ganglia pathways have been linked to movement implementation. There is the classical gating hypothesis (whereby the basal ganglia release a planned action). More recently, a vigor hypothesis has also been presented (where the basal ganglia control the vigor of executed actions). Given that STN stimulation here modulates movement times, I'm surprised there isn't more of a focus in the Introduction and Discussion on the movement vigor literature and how the results here may link up to them.

4. Clarity

There are many places where secondary analyses or an incomplete discussion distracts from the main story. This causes the reader to lose focus on the main narrative.

Examples of this include:

- Focus on Gamma & Theta power effects even though these are not used for the rest of the analyses.
- There is a clear influence of both task and STN stimulation impacting the decision threshold, a decision making parameter. In fact the decision threshold takes central focus in most of the paper. But then the conclusion shifts to saying that the results are inconsistent with a decision model (which includes the threshold). This pivot appears to come out of the blue.
- The effects of Speed vs. Accuracy instruction conditions all but disappear after the initial behavioral analysis. Yet the critical STN stimulation effects are on decision and movement speeds. Why are the STN stimulation (and beta power) effects not interacting with task manipulations that impact behavior in the same way?

MINOR COMMENTS

- Figure 2C isn't clear and is very hard to read.

- What is the value of plotting the posterior probability distributions in Figures 3B,C,E,F and 6B,E? While I am a fan of the Bayesian approach to understanding parameters, these plots take up a lot of space that could be revealed by simply reporting the confidence intervals in the text or a table.

- The color choice in Figure 5B makes it very hard to see what is a DBS_RT vs. DBS_MT stimulation effect. Instead of changing color hue, can these be change to different colors altogether?

- The term "IA" is used to as a shorthand for interaction, but this term isn't defined until deep into the methods at the end.

- line 277: the abbreviations "DBS_RT, DBS_MT, and DBS_OFF" are introduced with no context. The reader has to jump to the methods to understand what they are referring to.

- There is a constant jump from using cluster corrected p-values (to adjust for multiple comparisons) and one-tailed p-values. Why is one chosen for some and not others?

Signed: Timothy Verstynen

Point-by-point reply to reviewer comments:

Reviewer #1 (Remarks to the Author):

The authors speculate that the role of STN in decision-making is to control the release of a motor response. And to disentangle the cognitive decision-making and movement control, they designed the bilateral and unilateral DBS stimulation paradigm. The hypothesis is validated in a causal manner through the concrete evidence that only contralateral hand reaction time was affected. The manuscript is well-written and completes fancy computational modeling and data analyses. Especially, the artifact correction methods used for recovery of beta power modulation during stimulation trials are elegant.

Though it is honest to say that this study is well done and provides new knowledge, I hold major concerns regarding the design methods and interpretation of results. These major concerns, together with some minor ones, are listed below.

Reply: We thank the reviewer for the thoughtful comments, which have prompted us to thoroughly revise our manuscript. We hope that we have addressed all concerns as outlined below.

Major concerns on conceptual design of this study.

Comment 1: *1. Based on previous correlative evidence 10-12, the authorship hypothesized that the correlates and causal contributions of STN to deliberation precedes its effect on movement speed. A major issue here is that the too roughly-named 'deliberation' must be a complex structure containing many processes which may influence both decision accuracy and reaction time/speed. It seems that the authors overweighted speed but relatively devalue the accuracy. Their distinction should be explicitly informed and the focus of the whole study should be clearly stated.*

Reply: The main focus of this study were the correlates of decision and movement speed in the subthalamic nucleus (STN). While movement speed is well defined and can easily be measured, as the reviewer points out deliberation is more complex and involves different processes. For example, a person could have long reaction times, because noisy sensory evidence makes the task difficult or because they are very cautious and require a lot of evidence before committing to a choice. One way to disentangle these different latent processes is through modelling of the behavioral data with sequential sampling models, such as the drift diffusion model here. Rather than only analyzing reaction times or accuracy rates separately, these models can take both into account. In particular, noisy sensory evidence (task difficulty) corresponds to changes in the drift rate parameter (see figure 2A in the manuscript). Low drift rates result in long reaction times and more errors and these errors are typically slow. On the other hand, cautiousness corresponds to the decision threshold. High decision thresholds will also lead to longer reaction times, but these will be more likely to be correct (because more evidence has been accumulated) and when errors are caused by lower decision thresholds these will be fast (the accumulation process had been terminated prematurely). We here used a moving dots paradigm, which has been used very commonly for drift diffusion modeling in particular for the speed-accuracy trade-off (SAT). One putative disadvantage of this classical paradigm is that here the SAT mainly affects reaction times rather than accuracy rates as pointed out by Ratcliff & McKoon¹ and which we also have observed in our previous studies^{2,3}. However, by taking into account the reaction times distribution of correct and incorrect

responses the model can differentiate between changes in drift rate and decision thresholds (as well as the non-decision time) and thus make ‘decision speed’ less ambiguous. Importantly, we also found that errors were committed disproportionately faster after speed compared to accuracy instructions (manuscript page 5, see also comment 18), a telltale sign of reduced decision thresholds (if errors were caused by lower drift rates they should be committed relatively slow). We have made this more clear in the revised version of the manuscript (pages 3&5, all changes are marked in red). Throughout the paper we have also provided more tests regarding accuracy and provide a limitations section outlining shortcomings of the current paradigm and how this might be addressed in the future (page 18).

Comment 2: *2.Related to the above issue of accuracy, a critical finding is that though PD patients had lower accuracy rates compared to healthy people accuracy rates were not significantly different between Speed and Accuracy instructions. This at least partially contradicts the main goal of the study which aims to evaluate to what extent decision and movement processes are controlled by common or separate mechanisms. Since accuracy rate is a direct and critical measure for decision-making, the authors may have mainly measured movement-related processes but largely failed to measure decision-making processes (e.g., general cautiousness). Unfortunately, this pitfall in paradigm design may have rendered the global effect, if any, just undetectable.*

Reply: This comment is closely related to comment 1. Indeed, accuracy rates are less sensitive to this SAT paradigm than reaction times, but by considering the reaction time distribution of correct and incorrect responses it can be inferred whether the SAT instructions affected the decision threshold¹⁻³, which was confirmed by our modelling analysis (pages 6-8, see also reply to comment 1). Our previous interpretation that some of the mechanisms with which STN controls decision speed might be related to movement control mainly relied on the unilateral stimulation analysis but after careful reconsideration and in accordance with comments 1-3 by reviewer 2, we have toned down this interpretation, since it cannot be inferred unambiguously from the results (pages 17-18).

Comment 3: *3.Previous evidence linking STN to the computation of ‘higher-order’ decision-making, especially to the agent's general level of cautiousness, should be described in more detail. For example, to what extent a higher-order decision structure must exist globally? Is it possible that a from low to high hierarchy indeed exists but independently works for each hemisphere? These issues should be clarified and discussed more thoroughly.*

Reply: We agree with the reviewer that the results would be compatible with a hierarchical set of controllers where the basal ganglia reflect a lower level of hierarchy (compared to cortex) that controls decision thresholds of each hemisphere separately. We have thoroughly revised the abstract, introduction (page 4) and discussion (pages 17-18) to clarify these issues.

Comment 4: *4. In addition, I note that this study is similar to the Current Biology. 2018 paper regarding the paradigm used, the analyzing pipeline, and some of the results reported. The authors may elaborate on the common points and differences between the two studies, otherwise the novelty of this paper can be considerably reduced.*

Reply: The current study is novel compared to our previous studies in several regards. First, in our previous studies we used simple button presses to indicate a response^{2,3}, which does not allow analyzing movement speed. Since decision and movement speed are correlated this could

entail that any STN correlates of reaction times are mainly driven by changes in movement, especially since STN beta activity has classically been related to movement control. We could here demonstrate that STN control of decision and movement speed can be disambiguated and are controlled independently in the STN. Second, in our 2018 study² stimulation was only given bilaterally, which did not allow us to assess differential modulation of decision thresholds by the contra- and ipsilateral basal ganglia. Here, we demonstrate that, similarly to its effects on movement, STN controls decision speed for each hemisphere, which has important implications for neurobiological models of decision-making. Third, in the 2018 study² stimulation was triggered by STN beta activity and turned ON when beta activity was high. This makes it ambiguous whether the stimulation effects might mainly be driven by high beta power. We could resolve this here by applying random (i.e. non-beta triggered) stimulation. This could also demonstrate that stimulation-induced reduction of beta power does not necessitate high beta levels at the onset of stimulation. Fourth, the by-trial manipulation in our previous studies was task difficulty, while SAT was kept constant within blocks^{2,3}. Thus, our current results extend the previous findings by showing that dynamic STN control of decision thresholds generalize beyond task difficulty / conflict. Together, this study significantly extends our knowledge about basal ganglia control of decision and movement speed. We have revised the paper to make this more clear (pages 17-18).

Major concerns on interpretation for results.

Comment 5: *1. An interesting and important finding is that ‘at the single trial level longer reaction times were related to faster movements and this relationship did not differ depending on group or instructions’. I wonder what psychological mechanism underlies this negative correlation. One possibility is that longer reaction time affords a better preparation for the subsequent movement, providing stronger forces and thus faster speeds. However, this possibility is challenged by findings in beta power activity, which show that the beta band activities, though temporally separates decision and movement time in different window, are statistically independent from each other. Currently I do not see a clear path linking above findings together.*

Reply: There are several possibilities why reaction and movement times show a negative correlation. As the reviewer points out it could be that longer reaction times allow better movement preparation (which may not exclusively relate to beta power changes). Another possibility is that decisions with higher confidence (since more evidence has been accumulated) are indicated by faster movement, while people move slower when they are uncertain⁴. Since in our study movement times correlated with decision thresholds (i.e. movements were faster if more evidence had been accumulated), but not non-decision time (which incorporates movement preparation) this supports the latter. However, here we cannot directly infer movement preparation times and thus this inference should be taken with caution. The observed beta power changes in the STN have the same directionality, i.e., if beta decreases more strongly around the cue reaction times are shorter and if beta decreases more strongly around the response, movement times are shorter. We did not find a correlation between these two beta changes, but - if there was - this correlation should be positive, not negative. A simple explanation for this discrepancy might be that STN beta mediates the shift to lower thresholds and faster movement (in separate processing windows, see page 10 and figure 2D), but that it is not primarily involved in the inherent negative correlation between reaction and movement times (irrespective of instruction), which might be related to other spectral components or neural circuits outside the STN. We have included a short summary of these considerations in the revised manuscript (page 16).

Comment 6: 2. In line 278 and Figure 5B, the authors stated that ‘during DBSRT compared to DBSMT beta power was reduced from 20 ms precue to 160 ms postcue. Several points are unclear. First, the ratio of trials with DBS stimulation during both RT and MT is unshown. It is unclear to what extent DBSRT /DBSMT contains stimulation during RT/MT phase only. Second, it is a little puzzling to me why the effect of DBSRT appears mainly at postcue period (0-160ms). What is the interpretation? Third, in the middle panel of figure 5A stimulation windows have been highlighted for both RT and MT. As a direct validation, it is necessary to additionally show results for trials with stimulation in only these two windows.

Reply: The rationale for comparing DBS_{MT} vs DBS_{RT} trials was to match the trials as well as possible regarding recording, signal quality etc. However, we agree that this leads to some overlap, because in some DBS_{MT} trials stimulation was also applied in DBS_{RT} and vice versa. We have therefore now revised the analyses following the reviewer’s suggestion. Thus, we now compare STN beta power in the time window surrounding the stimulation effects contrasting all trials where stimulation was applied vs. all trials where no stimulation was applied in the DBS_{RT} and in the DBS_{MT} windows. This analysis shows that trials with stimulation in DBS_{RT} vs. No-DBS_{RT} (i.e., no stimulation in this time window) results in significantly lower beta power from 100 ms pre-cue to 60 ms post-cue. This also relates to the second point raised by the reviewer showing that with this revised analysis the beta reduction occurs in clear relation to DBS_{RT} rather than shifted to post-cue. For DBS_{MT} vs No-DBS_{MT} the analysis showed a DBS-related reduction of beta power from 320 to 460 ms postcue. I.e., very similarly to DBS_{RT}, we find that stimulation reduced beta power at around the time window where the DBS burst occurred. We have revised the manuscript (pages 13 & 38) and figure 5 accordingly.

Figure 5. Local field potential recordings during stimulation. **A.** Beta power is aligned to cue onset. **B.** Cue-aligned beta power for trials where stimulation was applied during DBS_{RT} (red) compared to trials where stimulation was not applied in DBS_{RT} (termed No-DBS_{RT}, black). Significant differences according to cluster-based permutation tests are marked by red bars with a *. **C.** Same as B, but for DBS_{MT} vs. No-DBS_{MT}. Note that the shown beta traces are not affected by movement-related beta power reductions and that the data was normalized to the mean across the whole recording (see methods). Shaded areas represent SEM.

Major concerns on methodological issues.

Comment 7: 1. The authors included both correct and incorrect trials into analysis. If results here are true, they should remain if only correct trials are left and incorrect trials removed. Though this will sacrifice 25%~35% incorrect trials (reference to 65%~75% accuracy rate reported in line 94), I suggest the authors rerun the analysis following this strategy.

Theoretically, incorrect trials are more susceptible to noise and this further explains why incorrect trials should be excluded.

Reply: As pointed out in reply to comment 1, for drift diffusion modelling and inferences on the latent parameters underlying the observed changes in decision speed, errors are vital. If all trials were correct, we could not infer whether decision speed was due to changes in evidence accumulation (drift rate) or response cautiousness (decision threshold). This is why we used a task that evokes a relatively large amount of errors without being purely random, based on our previous studies^{2,3}. In other words, errors here do not constitute (unintended) noise, but are an intricate part of the data. However, we agree with the reviewer it is important to check to what extent the results would hold true when only considering correct trials (except from drift diffusion modelling, which cannot be conducted without including errors). Importantly, the reported results do not change regarding their significance when only using correct trials including the behavioral and LFP results (listed in supplementary table 3). Furthermore, the effects of DBS on reaction times remained significant when only using correct trials, both for the effect of bilateral and contralateral stimulation (pages 11&14).

Comment 8: *2. In terms of stimulation, I have several major concerns.*

The authors used an approach of dynamic stimulation analysis to compare behavior and LFP differences between ON-stimulation and OFF-stimulation states. By setting parameters of burst stimulation (mean duration 250 ms, mean interval 150 ms), at any given 100 ms time-window, stimulation was applied to ~50% of trials. A ramping strategy was used to prevent sudden increases or decreases of the stimulation voltage.

First, time points during ramping were defined as no-stimulation. I do not believe this is the correct way to do. The ramping procedure is slow, lasting 130-350 ms. They stated that during this period, stimulation was below the effective intensity. This may be correct for motor improvement (not even so for time points when stimulation intensity is reaching the plateau), but not necessarily for cognitive processing. Therefore, I would at least divide some part of the ramping time to the ON-stimulation group (though the appropriate proportion is unclear). Or, more strictly, discarding data points during ramping since they may “contaminate” both the ON-stimulation and OFF-stimulation periods.

Reply: We agree with the reviewer that it is difficult to define at what intensity stimulation is ‘effective’. Since the most clear outcome measure is the motor response, we here titrated stimulation carefully for each patient to achieve clinically effective stimulation without side effects. Burst duration and intervals were chosen to achieve 50% stimulation trials for each window and thus changing these parameters will strongly affect % stimulation trials. For this reason, it is not feasible to completely discard ramping from the OFF definition, since this will leave hardly any OFF trials (and very low intensities are unlikely to have effects). However, we agree that some of the ramping might have already had cognitive effects. We therefore ran additional analyses in which we incrementally increased how much of the ramping was included as ON stimulation and assessed to what extent this affected the effects of DBS on reaction times. We did this from 10 ms until 50 ms of ramping up and down (including the last 50 ms of ramping up and first 50 ms of ramping down leads to 100 ms longer ON stimulation bursts). When including 50 ms ramping up and down only ~20% of OFF trials remained for each window and, given that there were ~60 valid trials per person for the Speed and 60 trials for the Accuracy condition, this only left ~ 10 OFF trials for each condition, which is why we did not increase the included ramping further (page 35). Importantly, this analysis showed that the behavioral results were robust to these adjustments of the stimulation definition both for

bilateral and contralateral stimulation (see pages 11 & 14 and supplementary table 6). We have also included a limitations section in the revised manuscript discussing these issues and outlining ways to mitigate this in future studies (page 18).

Comment 9: *Second, the stimulation interval lasted on average 150 ms was defined as NO-stimulation. I doubt its rationality. This period better represents post-stimulation state rather than no-stimulation. It is clear that basal ganglia neurons cannot return to the pre-stimulation/no-stimulation state within a short 150 ms interval (instead, at least 2-3 s may need (see Science. 2021 paper)). Following this conception, all results based on these comparisons are actually revealing differences between with-stimulation and shortly-after stimulation (when neurons are still affected by the stimulation).*

Reply: We agree that the very fast cycling between ON and OFF stimulation in this study presumably did not result in a stable ‘no stimulation’ state during the intervals, which might rather constitute a ‘post-stimulation’ period. We have included this important insight in the revised version of the manuscript and discuss how this might be mitigated in future work (page 18). However, the contrast between ON and OFF stimulation / post-stimulation still provides support for a mechanistic role for the STN in controlling the speed of decisions and movement, and for this being independently achieved for each hemisphere at the STN level during adaptive behavior.

Comment 10: *Third, DBS-induced changes in accuracy rate should be reported for both bilateral and unilateral stimulations.*

Reply: Following the reviewer’s suggestion we have now also analyzed whether DBS effects on reaction times were accompanied by changes in accuracy rates. These effects (which were not significant) are now included in the revised results sections (pages 11 & 14). Of note, we have conducted drift diffusion modelling of the DBS effects which show that these are in line with changes in the decision threshold both for bilateral stimulation and contralateral stimulation (pages 11 & 14). Please see also replies to comments 1&2 regarding the limited sensitivity of the current paradigm to changes in accuracy rates.

Comment 11: *Fourth, I recommend a comparison between activity from the contralateral vs. ipsilateral STN contacts, to evaluate to what extent signals are intrinsically isolated or shared between hemispheres.*

Reply: We thank the reviewer for suggesting this helpful analysis. Following the reviewer’s suggestion we now also analyzed to what extent STN beta activity showed any lateralization. While the modulation of STN activity by the cue and movement looks highly similar in both hemispheres, beta power was overall lower in the contralateral compared to the ipsilateral STN (cluster-based permutations, see pages 8 & 9 and supplementary figure 2). This finding fits in nicely with the lateralization of effects from the stimulation analysis.

Supplementary figure 2. Lateralization of subthalamic beta power. **A.** Changes in beta power aligned to the cue. **B.** Changes in beta power aligned to the movement. Black bars with a * indicate time windows where beta power was lower in contralateral (black) compared to ipsilateral (grey) subthalamic nucleus according to cluster-based permutation tests ($P_{\text{cluster}} < 0.05$). Comparing $\text{Beta}_{\text{move}}$ and Beta_{cue} (see text) between hemispheres showed that $\text{Beta}_{\text{move}}$ was significantly lower contralateral vs. ipsilateral (CI [-0.168;-0.024], $P = 0.014$) while there was no significant difference in Beta_{cue} (CI [-0.064:+0.061], $P = 0.959$). Shaded areas represent SEM.

Comment 12: *Fifth, different with bilateral stimulation analysis (Figure 4D and 4F) the authors did not subdivide trials into Speed vs. Accuracy for unilateral stimulation (as explained by authors, in order to assure a sufficient number of trials). Though the authors explained that this was to assure a sufficient number of trials (line 803), I believe additional analyses are required to evaluate the behavioral modulation effect of unilateral stimulation for accuracy and speed trials, respectively. This will help to tell whether the effect of Instruction (Speed vs. Accuracy) is dependent on bilateral stimulation or not.*

Reply: Following the reviewer's suggestion we have now further subdivided these trials into speed vs. accuracy despite the paucity of trials. Interestingly this shows that contralateral stimulation did not differently affect reaction times in speed vs. accuracy trials and that it indeed reduced reaction times consistently in both conditions (8/8 patients in accuracy condition, 7/8 patients in speed condition). As the reviewer points out this further elucidates differences between unilateral and bilateral stimulation indicating that context-dependent decision thresholds adjustment might require the controlled activation of bilateral STN in line with a 'lower-hierarchy' function of the STN (pages 14 & 17-18).

Comment 13: *3.As stated in line 493, for each trial the beginning of the force exertion was marked manually where a clear offset from baseline was visible at a fixed temporal resolution (1 s/inch) and blinded to trial type. I wonder why the beginning point has to be marked manually. If possible, I recommend they determine these starting points using algorithms. This would make it more replicable for other researchers to follow.*

Reply: We agree that for replicability an automatic algorithm would be best. However, since this study involves patients with Parkinson's disease the force measures are sometimes affected by some involuntary movement e.g. action tremor, which is often subtle, but can trigger incorrect detection of movement onset (an example of correct and incorrect automatic movement detection is shown in figure R1 below).

successful:

detected movement onset

unsuccessful:

detected movement onset

Figure R1. X-axis shows time in ms (1000 ms corresponds to the movement detected by PsychoPy), y-axis is force (au). The arrow and vertical line indicate movement onset as detected by the automatic algorithm, which in the right panel was triggered by a small variation in force, presumably due to subtle action tremor.

Therefore, in a significant proportion of trials it was necessary to set the onset manually. To avoid any putative group bias (manual adjustments are mainly necessary in patients, while the data from healthy people usually does not require this) we opted to always set the onset manually blinded to trial-type. However, we have now also tested whether automatic detection of movement onset (threshold for movement onset defined as > 5 times the standard deviation of the baseline period, without manual alterations) might alter any of the results. Importantly, both methods lead to similar movement times (Pearson correlation between the two measures is $\rho = 0.637$, $P < 0.0001$) and consequently highly similar behavioral results (none of the key results are affected). We have now also included a description of the procedure of the automatic detection and their similar outcomes in the revised version (page 22).

Comment 14: 4. In Figure 3B, the authors calculated beta power decrease as the differential mean power between window 300-100 ms precue and window 320-400ms postcue. How these two windows are determined? To avoid p-hacking, a more rigorous and better solution is to let data say by itself. Similarly, why the window for Theta power analysis in Figure 4C switched to 0-700ms postcue?

Reply: In order to conduct regression analyses with behavior and for drift diffusion modelling with neural data we extracted several time-frequency windows of interest. For Beta_{cue} , we chose a baseline where beta power before the cue was stable. The rationale for using the window from 320-400 ms postcue is that here beta power ‘plateaus’ after the initial cue-induced decrease and before the later movement-related beta decrease (moving this window further away from the cue would require more and more trials to be excluded to avoid a ‘spill-

over' between the cue- and movement-aligned beta change). The reduction of beta to 400ms postcue is very consistent and we have found the same window (and used beta power to this timepoint for regression analyses with reaction times and decision thresholds) in our previous papers^{2, 3}. We describe this rationale more clearly in the revised manuscript (page 31). However, we agree that picking these exact windows might seem arbitrary and could be interpreted as p-hacking. We have therefore now included control analyses, where we move the baseline window (200 ms duration) from 500 ms precue until the cue (the postcue window was not moved because of the reasons mentioned above). Importantly, beta power robustly predicts RT irrespective of the exact window chosen, i.e. the results do not depend on using this specific window (see page 9 and supplementary table 5).

For theta, a 750 ms window between cue and movement was chosen based on our previous study³. Due to comments from both reviewers we have moved theta changes to the supplementary material, since it does not provide key findings (see also comment 21 and comment 7 of reviewer 2).

Comment 15: *5.In most cases, the authors only reported CrI to evaluate the statistical significance. To help readers better evaluate the results, I recommend authors show more comprehensive measures, including but may be not restricted to correlation coefficient, regression coefficient degree of freedom and p-values, when necessary for their key findings.*

Reply: We mainly reported the credible interval (CrI), because Bayesian statistics are not based on classical p-values, but on the posterior distribution of parameter estimates. The most frequently used interval in Bayesian statistics is the CrI, which is closely related to the frequentist confidence interval (CI) and intuitive to understand. Significance can simply be inferred by considering whether 95% of the CrI overlaps with 0. Since Bayesian statistics are not as widely used as frequentist statistics we had also included a table with the CI. Following the reviewer's suggestion we now also include additional frequentist parameters in the revised version of supplementary tables 2-4 (of note we still report the CI of the regression coefficient rather than its mean because it more directly corresponds to the CrI).

Minor issues

Comment 16: *1.In Processing of STN LFPs, Methods, according to the authors, for directional leads, bipolar rereference was made between "the most dorsal omnidirectional contact and the neighbouring three dorsal directional contacts, between the three dorsal and corresponding three ventral directional contacts, as well as the three ventral directional contacts and neighbouring most ventral omnidirectional contact". Following this conception, 9 rather than 8 bipolar channels would be produced. Please check this.*

Reply: We are sorry for this mistake, which has been corrected (page 28).

Comment 17: *2.In Processing of STN LFPs, Methods, the authors confirmed functional localizer approach by further conducting lead localization analysis. However, imaging data were not available in three subjects. Do the authors rely solely on the functional localizer results (which is also accepted), or do they conduct other analyses on these three subjects to confirm that the lead is within the dorsal STN? Please report.*

Reply: For these three subjects only the functional localizer was conducted. Another indirect measure is the clinical response to DBS, which would be attenuated or complicated by side effects if the stimulation electrode was outside the STN. We provide this information in the revised manuscript (pages 29-30).

Comment 18: 3. *In line 99, Accuracy rates...but participants committed errors disproportionately faster after Speed vs. Accuracy instruction as indicated by a significant Instruction*accuracy IA on reaction times (CrI [+0.042:+0.223])... I wonder how to interpret the result and what psychological mechanism underlies this positive correlation. In addition, is it necessary to study the effect of the accuracy on the movement time as well?*

Reply: Please see our detailed replies to comments 1 and 2 above regarding the reaction time distribution of correct and incorrect responses and their indication for inference on decision threshold adjustments. We have now also analyzed effect of response accuracy on movement time, which did not show any significant effects presumably since movement times were much shorter than reaction times (page 6 and supplementary table 2).

Comment 19: 4.*In line 501, Single trial values of reaction times, movement times and peak force of each 502 subject j were log-transformed (because of their skewed distribution). I think it's better to plot the distribution of reaction times, movement times and peak force, and illustrate the parameters of the skewed distributions.*

Reply: We log-transformed the data, because in the Bayesian statistical models data was assumed to be drawn from a normal distribution, but reaction times, movement times and peak force had a heavy tail (which is very common for these measures). Following the reviewer's suggestion we have included an additional figure with the raw (and log-transformed) data for transparency (suppl. figure 9) and have also assessed whether the significant results hold when using raw rather than logged-transformed data. Importantly all key results, both regarding behavior and LFPs, remain significant (listed in suppl. table 4).

Supplementary figure 9: Distribution of raw (left column) and log-transformed (right column) behavioral data. Raw movement times, reaction times and peak force had a heavy tail. Since the Bayesian model assumed a normal distribution this data was log transformed, but the significant key results did not change when omitting this step (listed in supplementary table 4).

Comment 20: *5.typo errors exist. For example, in line 281 figure 5B was mistakenly spelled as figure 5F. NOTE that several more similar errors have been noticed but not listed here.*

Reply: We apologize for any errors and have carefully re-examined the paper for any typos during the revision.

Comment 21: *6.To keep the whole study concise, I suggest placing theta and gamma-related results in the supplementary material.*

Reply: We agree with the reviewer and have placed these results in the supplemental material (supplementary figure 3).

Reviewer #2 (Remarks to the Author):

This incredibly interesting manuscript describes a rare set of behavioral, electrophysiological,

and stimulation experiments in humans, focused on the subthalamic nucleus' (STN's) role in regulating decision and motor states. The authors had participants perform a simple variant of the dot motion task, indicating the left/right motion direction of the dots by squeezing a dynamometer in either hand. Participants were told via visually displayed cues whether to focus on speed or accuracy of their responses, which were then modeled using the drift diffusion model (DDM). Patients with deep brain stimulation electrodes implanted in the STN were tested alongside non-patient controls.

The authors first confirm prior reports that speed constraints on decision time relate to a compression of the boundary height from the DDM, which also appears to correlate with movement times. Next, they show that beta frequencies in the STN drop after the imperative cue and rebound after the response has been made, indicating a role in the decision and movement. Then, and most importantly, they found that there were two critical windows for stimulation that impacted task performance: 1) an early (10-180ms) period before the cue that decreased reaction times, and 2) a late (330-460ms) period that impacted movement times. The authors then show how stimulation in each of these windows disrupted STN beta responses. Finally, the authors show how the effects of stimulation on behavior and STN beta power are lateralized to the side of the brain contralateral to the responding hand. The authors conclude from this evidence that the STN is more likely to be involved in implementation of motor actions (specifically by withholding the prepared action), rather than decision-making.

All in all, I thought this was a very unique and interesting study that can contribute a lot to our understanding of the role of basal ganglia nuclei in decision making and motor implementation in humans. However, there are some inferential and narrative problems that make the power of the findings hard to see as a reader.

Reply: We thank the reviewer for the positive evaluation and very helpful comments, which helped us to improve the manuscript and our interpretation of the results. We hope that we have answered all concerns as detailed below.

MAJOR CONCERNS

Comment 1: *1. Inference.*

If I understand this correctly, the authors conclude that the STN's role in this task is more motoric (i.e., withholding a planned action) than cognitive (i.e., computing the decision threshold) based primarily on 3 bits of evidence.

- Beta power in the STN did not appear to be modulated by speed-vs-accuracy cue instructions (though other frequency spectrums in the STN do).*
- Stimulation decreased reaction times in the pre-cue window.*
- Stimulation decreased movement times in the post-cue window.*

I have three problems with this. First, the contrastive framing of the two alternative hypotheses (computing decision threshold vs. gating the motor response) is itself a sort of strawman of the competing theories on STN/basal ganglia literature on their roles during response selection. While many have pointed out a role for the STN (or broader indirect pathway) in modulating the decision threshold, many computational theorists posit that the threshold itself is either implemented downstream or as a distributed computation across multiple regions. On the other hand, the movement vigor hypothesis (a variant of the implementation hypothesis presented in

this paper), states that the movement speed is regulated by a competition between the direct and indirect pathways (a similar mechanism has been postulated for modulating the drift-rate). Both of these more nuanced ways of understanding the computation of the STN make this experiment a less severe test (to borrow Popper's term) of each hypothesis.

Reply: We agree with the reviewer that the results of this study do not unambiguously support our previous interpretation and have thoroughly revised the manuscript. In particular, we agree that our results are in line with both proposed models of the basal ganglia supporting decision-making by determining when to commit to a choice and supporting movement control by determining how vigorously to perform a movement. Since the following comments are closely related to this, please see also replies to comment 2&3. The corresponding revisions of the manuscript are marked in red (abstract, pages 3, 17 & 18).

Of note, STN beta power was modulated by speed vs. accuracy instructions in that Beta_{cue} decreased more strongly after speed instructions and the relationship between $\text{Beta}_{\text{move}}$ and movement times depended on the type of instruction.

Comment 2: *Second, and perhaps more importantly, even within the narrow framing of the contrastive hypothesis that the authors adopted here, I am not sure that the evidence itself is strong enough to warrant the authors conclusion. Yes, reaction times appear to speed up when STN stimulation happens before the cue. But I don't see any analysis on the accuracy of responses. If response accuracy is also impacted by pre-cue stimulation, then this indicates a disruption of the decision process itself, rather than the motor response. Without this analysis it is hard to fully understand what is happening as a result of STN stimulation.*

Reply: Since this comment is closely related to comment 1 of reviewer 1, we have copied parts of our response here.

‘We here used a moving dots paradigm, which has been used vastly for drift diffusion modeling in particular for the speed-accuracy trade-off (SAT). One putative disadvantage of this classical paradigm is that the SAT mainly affects reaction times rather than accuracy rates as pointed out by Ratcliff & McKoon¹ and which we also have observed in our previous studies^{2, 3}. However, by taking into account the reaction times distribution of correct and incorrect responses the model can differentiate between changes in drift rate and decision thresholds (as well as the non-decision time) and thus make ‘decision speed’ less ambiguous. Importantly, we also found that errors were committed disproportionately faster after speed compared to accuracy instructions (manuscript page 5), a telltale sign of reduced decision thresholds (if errors were caused by lower drift rates they should be committed relatively slow). We have made this more clear in the revised version of the manuscript (pages 3 & 5). Throughout the paper we have also provided more tests regarding accuracy and provide a limitations section outlining shortcomings of the current paradigm and how this might be addressed in the future (page 18).’

Of note, since in this task the motor response terminates the decision process a mere ‘motor process’ could also result in increased error rates. We agree, however, that the results do not strictly allow the conclusion from the previous manuscript version and have therefore thoroughly revised the paper (see reply to comment 1).

Comment 3: *Finally, and related to the previous point, it is not clear why both hypotheses cannot be true. STN stimulation clearly had two windows of efficacy separated by several hundred milliseconds. Within the framing of the hypotheses presented by the authors, it seems possible that two phases of signal come through the STN: an early phase related to the decision*

and a later phase related to the execution. I am not certain how the current results cannot rule out this third hypothesis. Why was accuracy excluded as a primary outcome measure of interest, particularly when it's relevant to the DDM?

Reply: We thank the reviewer for this important insight, which has prompted us to thoroughly revise our paper (abstract, pages 3, 17 & 18). We agree that our results are in line with the basal ganglia supporting decision-making by determining when to commit to a choice and supporting movement control by determining how vigorously to perform a movement.

As stated in the previous replies we have now included more analyses of accuracy rates, but acknowledge that the current experimental paradigm is not particularly suited for this outcome measure (see above).

Comment 4: 2. Analysis

I really liked the careful DDM analysis used in this paper. However, I am a bit confused as to why the bias term (z) was not included as a free parameter in the early model evaluations. This seems like an incomplete vetting of the effects of task (and stimulation) on the underlying decision parameters. While I believe the results shown here focused on the decision threshold, particularly since they align with many previous reports, it is still possible that including bias (z) could change some of the results shown here.

Reply: We thank the reviewer for this helpful comment. The reason for not including this parameter was that, in our experience with experiments with relatively low trial counts as often is the case for patient data, the DDM has convergence issues the more complex the models become. Therefore, we conducted a relatively complex model first and then used the reduced model for the next analyses. However, we agree that in this paradigm the bias parameter (z) indeed might be relevant, in particular since we focus on decision threshold computations. We have therefore included this term in the initial model estimation, which showed that it was not different from 0.5 (i.e. centered between response options) according to its 95% CrI. For all HDDM analyses we now also ran additional models including the bias term. This did not alter any of the significant results reported in the study. To keep the paper comprehensive we still report the original results, but now explain the bias term and report that key results were not different when including this parameter in the models (pages 7 & 26).

Comment 5: 3. Vigor

There are two ways that the basal ganglia pathways have been linked to movement implementation. There is the classical gating hypothesis (whereby the basal ganglia release a planned action). More recently, a vigor hypothesis has also been presented (where the basal ganglia control the vigor of executed actions). Given that STN stimulation here modulates movement times, I'm surprised there isn't more of a focus in the Introduction and Discussion on the movement vigor literature and how the results here may link up to them.

Reply: We thank the reviewer for bringing up this important point and have revised both introduction and discussion so as to incorporate this hypothesis (pages 3 & 17).

Comment 6: 4. Clarity

There are many places where secondary analyses or an incomplete discussion distracts from the main story. This causes the reader to lose focus on the main narrative.

Reply: We apologize for this and have revised the manuscript according to the reviewer's suggestions.

Comment 7: *Examples of this include:*

- *Focus on Gamma & Theta power effects even though these are not used for the rest of the analyses.*

Reply: These are now only briefly mentioned in the main text, but have mainly been moved to supplementary figure 3 (see also comment 21 of reviewer 1).

Comment 8: *There is a clear influence of both task and STN stimulation impacting the decision threshold, a decision making parameter. In fact the decision threshold takes central focus in most of the paper. But then the conclusion shifts to saying that the results are inconsistent with a decision model (which includes the threshold). This pivot appears to come out of the blue.*

Reply: We agree and have revised the paper accordingly (pages 17-18, see also our replies above).

Comment 9: - *The effects of Speed vs. Accuracy instruction conditions all but disappear after the initial behavioral analysis. Yet the critical STN stimulation effects are on decision and movement speeds. Why are the STN stimulation (and beta power) effects not interacting with task manipulations that impact behavior in the same way?*

Reply: The relationship of beta power with, respectively, reaction and movement times, both depend on task instructions. Beta_{cue} decreases more after Speed instructions and is related to shorter reaction times and lower decision thresholds (figure 3B). $\text{Beta}_{\text{move}}$ is related to movement speed only after Speed, but not after Accuracy instructions (figure 3D). Thus, beta power modulations accompany speed-accuracy adjustments. Bilateral STN stimulation affected reaction times and decision thresholds depending on speed vs. accuracy instructions, since it only reduced reaction times and decision thresholds in the speed condition (figure 4 C&D). An exception to this context-dependence is that contralateral stimulation reduced reaction times irrespective of condition (see also comment 12 of reviewer 1), which we now discuss in the revised manuscript (page 17).

MINOR COMMENTS

Comment 10: - *Figure 2C isn't clear and is very hard to read.*

Reply: We apologize and have plotted this in its own figure to make it larger and easier to read (suppl. figure 1).

Supplementary figure 1. Quantile probability plots showing the observed (indicated by a cross) and predicted (indicated by ellipses) cumulative probabilities of reaction times separately for the two groups (Parkinson’s disease (PD) & healthy controls (HC)) and Instructions (speed & accuracy). Correct choices are shown in black, incorrect choices in red. The width of the circles reflects prediction uncertainty (standard deviation of the posterior predictive distribution).

Comment 11: - What is the value of plotting the posterior probability distributions in Figures 3B,C,E,F and 6B,E? While I am a fan of the Bayesian approach to understanding parameters, these plots take up a lot of space that could be revealed by simply reporting the confidence intervals in the text or a table.

Reply: We included these since in our view they give a good intuition about effect size and uncertainty. We have now moved the results pertaining to theta and gamma to the supplementals, so that revised figure 3 hopefully now is more comprehensive.

Figure 3. Local field potential recordings from subthalamic nucleus. **A.** Grand average for spectra aligned to the moving dots cue. **B.** Results for cue-aligned changes in beta (~13-30 Hz) power ($Beta_{cue}$) measured as change in beta power from precue (300-100 ms precue) to postcue (320-400 ms postcue). **C.** Grand average for spectra aligned to movement onset. **D.** Results for movement-related (0-300 ms post-movement) changes in beta power ($Beta_{move}$). In B and D shaded areas around %

power change (green: speed; black: accuracy) represent SEM and grey vertical boxed indicate time windows from which power was extracted. Green and black vertical lines in B indicate mean reaction time.

Comment 12: - *The color choice in Figure 5B makes it very hard to see what is a DBS_RT vs. DBS_MT stimulation effect. Instead of changing color hue, can these be change to different colors altogether?*

Reply: We now separate the two effects in different panels and use red (vs. black) for both (revised figure 5, see also comment 6 of reviewer 1).

Figure 5. Local field potential recordings during stimulation. **A.** Beta power is aligned to cue onset. **B.** Cue-aligned beta power for trials where stimulation was applied during DBS_{RT} (red) compared to trials where stimulation was not applied in DBS_{RT} (termed No-DBS_{RT}, black). Significant differences according to cluster-based permutation tests are marked by red bars with a *. **C.** Same as B, but for DBS_{MT} vs. No-DBS_{MT}. Note that the shown beta traces are not affected by movement-related beta power reductions and that the data was normalized to the mean across the whole recording (see methods). Shaded areas represent SEM.

Comment 13: - *The term “IA” is used to as a shorthand for interaction, but this term isn’t defined until deep into the methods at the end.*

Reply: The abbreviation IA is introduced in the introduction (page 5).

Comment 14: - *line 277: the abbreviations “DBS_RT, DBS_MT, and DBS_OFF” are introduced with no context. The reader has to jump to the methods to understand what they are referring to.*

Reply: We apologize for this and have revised the paper accordingly (pages 11 & 12).

Comment 15: - *There is a constant jump from using cluster corrected p-values (to adjust for multiple comparisons) and one-tailed p-values. Why is one chosen for some and not others?*

Reply: For frequentist analyses we used cluster-based permutations whenever the data included a continuous measure over time (e.g. DBS sliding window analyses). For post-hoc tests (e.g. does DBS in the window where it affects reaction times also affect accuracy rates or post-hoc tests after interactions) we used simple t-tests. However, we agree that using one-tailed p-values often seems arbitrary and could be interpreted as p-hacking, which is why we removed any one-tailed results that did not have a clear a-priori hypothesis. Throughout the

frequentist analysis there is now only one one-tailed test, which compares the timing-specific effects of contralateral vs. ipsilateral stimulation (post-hoc after the cluster-based permutation). We believe that this is justified, since here the direction of expected effects was clear (it should be stronger after contralateral compared to ipsilateral DBS, since only the former was significant in the initial test over multiple time windows). We now state this more explicitly in both the methods (page 25) and results section (page 14).

Signed: Timothy Verstynen

References:

1. Ratcliff, R. & McKoon, G. The diffusion decision model: theory and data for two-choice decision tasks. *Neural Comput* **20**, 873-922 (2008).
2. Herz, D.M., *et al.* Mechanisms Underlying Decision-Making as Revealed by Deep-Brain Stimulation in Patients with Parkinson's Disease. *Curr Biol* **28**, 1169-1178 e1166 (2018).
3. Herz, D.M., *et al.* Distinct mechanisms mediate speed-accuracy adjustments in cortico-subthalamic networks. *Elife* **6** (2017).
4. Seideman, J.A., Stanford, T.R. & Salinas, E. Saccade metrics reflect decision-making dynamics during urgent choices. *Nat Commun* **9**, 2907 (2018).

REVIEWERS' COMMENTS

Reviewer #1 (Remarks to the Author):

The authors have done a great job.

Reviewer #2 (Remarks to the Author):

The authors have done an excellent job addressing all of my points. I have no further comments on this manuscript.

-Tim Verstynen

Point-by-point reply to reviewer comments:

Reviewer #1 (Remarks to the Author):

The authors have done a great job.

Reply: We thank the reviewer for the positive evaluation and the thoughtful comments to the previous manuscript version, which have significantly improved the paper.

Reviewer #2 (Remarks to the Author):

The authors have done an excellent job addressing all of my points. I have no further comments on this manuscript.

-Tim Verstynen

Reply: Thank you Dr. Verstynen for the very helpful comments to the previous version, which we believe have greatly improved the paper, and the positive evaluation.